# Derivatives and residual distribution of regularized M-estimators with application to adaptive tuning

## Abstract

This paper studies M-estimators with gradient-Lipschitz loss function regularized with convex penalty in linear models with Gaussian design matrix and arbitrary noise distribution. A practical example is the robust M-estimator constructed with the Huber loss and the Elastic-Net penalty and the noise distribution has heavy-tails. Our main contributions are three-fold. (i) We provide general formulae for the derivatives of regularized M-estimators $\widehat{\boldsymbol{\beta}}(\boldsymbol{y}, \boldsymbol{X})$ where differentiation is taken with respect to both $\boldsymbol{y}$ and $\boldsymbol{X}$; this reveals a simple differentiability structure shared by all convex regularized M-estimators. (ii) Using these derivatives, we characterize the distribution of the residual $r_i = y_i - \boldsymbol{x}_i^\top \widehat{\boldsymbol{\beta}}$ in the intermediate high-dimensional regime where dimension and sample size are of the same order. (iii) Motivated by the distribution of the residuals, we propose a novel adaptive criterion to select tuning parameters of regularized M-estimators. The criterion approximates the out-of-sample error up to an additive constant independent of the estimator, so that minimizing the criterion provides a proxy for minimizing the out-of-sample error. The proposed adaptive criterion does not require the knowledge of the noise distribution or of the covariance of the design. Simulated data confirms the theoretical findings, regarding both the distribution of the residuals and the success of the criterion as a proxy of the out-of-sample error. Finally our results reveal new relationships between the derivatives of $\widehat{\boldsymbol{\beta}}(\boldsymbol{y}, \boldsymbol{X})$ and the effective degrees of freedom of the M-estimator, which are of independent interest.

## 1 Introduction

This paper studies properties of robust estimators in linear models $\boldsymbol{y} = \boldsymbol{X}\boldsymbol{\beta}^* + \boldsymbol{\varepsilon}$ with response $\boldsymbol{y} \in \mathbb{R}^n$, unknown regression vector $\boldsymbol{\beta}^*$ where $\boldsymbol{X}$ is a design matrix with $n$ rows $\boldsymbol{x}_1, ..., \boldsymbol{x}_n$, each row $x_i$ being a high-dimensional feature vector in $\mathbb{R}^p$ with covariance $\boldsymbol{\Sigma}$. Throughout, let $\widehat{\boldsymbol{\beta}} = \widehat{\boldsymbol{\beta}}(\boldsymbol{y}, \boldsymbol{X})$ be a regularized $M$-estimator given as a solution of the convex minimization problem

$$\widehat{\boldsymbol{\beta}}(\boldsymbol{y}, \boldsymbol{X}) = \mathrm{argmin}_{\boldsymbol{b} \in \mathbb{R}^p} \tfrac{1}{n} \sum_{i=1}^n \rho(y_i - \boldsymbol{x}_i^\top \boldsymbol{b}) + g(\boldsymbol{b}) \tag{1}$$

where $\rho : \mathbb{R} \to \mathbb{R}$ is a convex data-fitting loss function and $g : \mathbb{R}^p \to \mathbb{R}$ a convex penalty. We may write $\widehat{\boldsymbol{\beta}}_{\rho,g}(\boldsymbol{y}, \boldsymbol{X})$ for (1) to emphasize the dependence on the loss-penalty pair $(\rho, g)$; if the argument $(\boldsymbol{y}, \boldsymbol{X})$ is dropped then $\widehat{\boldsymbol{\beta}}$ is implicitly understood at the observed that $(\boldsymbol{y}, \boldsymbol{X})$. Typical examples of losses include the square loss $\rho(u) = u^2/2$, the Huber loss $H(u) = \int_0^{|u|} \min(1, t) dt$ or its scaled version $\rho = \Lambda^2 H(u/\Lambda)$ for some tuning parameter $\Lambda > 0$, while typical examples of penalty functions include the Elastic-Net $g(\boldsymbol{b}) = \lambda\|\boldsymbol{b}\|_1 + \mu\|\boldsymbol{b}\|^2/2$ for tuning parameters $\lambda, \mu \geq 0$.

The paper introduces the following criterion to select a loss-penalty pair $(\rho, g)$ with small out-of-sample error $\|\boldsymbol{\Sigma}^{1/2}(\widehat{\boldsymbol{\beta}} - \boldsymbol{\beta}^*)\|^2$: for a given set of candidate loss-penalty pairs $\{(\rho, g)\}$ and the

34 corresponding $M$-estimator $\widehat{\boldsymbol{\beta}}_{\rho,g}$ in (1), select the pair $(\rho, g)$ that minimizes the criterion

$$\text{Crit}(\rho, g) = \left\| \boldsymbol{r} + \frac{\hat{\mathsf{df}}}{\text{tr}[\boldsymbol{V}]} \psi(\boldsymbol{r}) \right\|^2 \text{ with } \begin{cases} \boldsymbol{r} = \boldsymbol{y} - \boldsymbol{X}\widehat{\boldsymbol{\beta}}_{\rho,g} & \in \mathbb{R}^n, \\ \hat{\mathsf{df}} = \text{tr}[\boldsymbol{X}(\partial/\partial \boldsymbol{y})\widehat{\boldsymbol{\beta}}_{\rho,g}] & \in \mathbb{R}, \\ \boldsymbol{V} = \text{diag}\{\psi'(\boldsymbol{r})\}(\boldsymbol{I}_n - \boldsymbol{X}(\partial/\partial \boldsymbol{y})\widehat{\boldsymbol{\beta}}_{\rho,g}) & \in \mathbb{R}^{n \times n} \end{cases} \quad (2)$$

35 where $\text{tr}[\cdot]$ is the trace, $\psi : \mathbb{R} \to \mathbb{R}$ is the derivative of $\rho$, $\psi'$ the derivative of $\psi$ and we extend $\psi$
36 and $\psi'$ to functions $\mathbb{R}^n \to \mathbb{R}^n$ by componentwise application of the univariate function of the same
37 symbol. Above, $(\partial/\partial \boldsymbol{y})\widehat{\boldsymbol{\beta}}_{\rho,g} \in \mathbb{R}^{p \times n}$ denotes the Jacobian of (1) with respect to $\boldsymbol{y}$ for $\boldsymbol{X}$ fixed,
38 at the observed data $(\boldsymbol{y}, \boldsymbol{X})$. As we will see while studying particular examples, for pairs $(\rho, g)$
39 commonly used in robust high-dimensional statistics such as the square loss, Huber loss with the
40 $\ell_1$-penalty or Elastic-Net penalty, the ratio $\hat{\mathsf{df}}/\text{tr}[\boldsymbol{V}]$ in (2) admits simple, closed-form expressions
41 and can be computed at a negligible computational cost once $\widehat{\boldsymbol{\beta}}_{\rho,g}(\boldsymbol{y}, \boldsymbol{X})$ itself has been computed.
42 The criterion (2) has an appealing adaptivity property: it does not require any knowledge of the noise
43 $\boldsymbol{\varepsilon}$ or its distribution, nor any knowledge of the covariance $\boldsymbol{\Sigma}$ of the design.

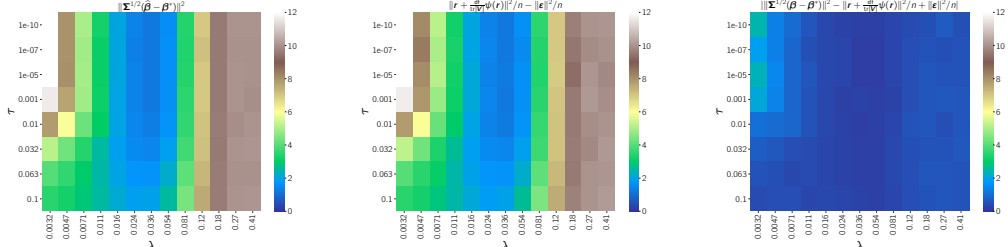

Figure 1: Heatmaps for $\|\boldsymbol{\Sigma}^{1/2}(\hat{\boldsymbol{\beta}} - \boldsymbol{\beta}^*)\|^2$, its approximation $\|\boldsymbol{r} + (\hat{\mathsf{df}}/\text{tr}[\boldsymbol{V}])\psi(\boldsymbol{r})\|^2/n - \|\boldsymbol{\varepsilon}\|^2/n$ and the approximation error $\|\|\boldsymbol{\Sigma}^{1/2}(\hat{\boldsymbol{\beta}} - \boldsymbol{\beta}^*)\|^2 - \|\boldsymbol{r} + (\hat{\mathsf{df}}/\text{tr}[\boldsymbol{V}])\psi(\boldsymbol{r})\|^2/n - \|\boldsymbol{\varepsilon}\|^2/n|$ for the Huber loss and Elastic-Net penalty on a grid of tuning parameters $(\lambda, \tau)$ where $\lambda \in [0.0032, 0.41]$ and $\tau \in [10^{-10}, 0.1]$. Each cell is the average over 100 repetitions. See Section 6 for more details.

## 1.1 Contributions

45 1. The end goal of paper is to provide theoretical justification and theoretical guarantees for the
46 criterion (2) in the high-dimensional regime where the ratio $p/n$ has a finite limit and $\boldsymbol{X}$ has
47 anisotropic Gaussian distribution. The theoretical results will justify the approximation

$$\left\| \boldsymbol{r} + (\hat{\mathsf{df}}/\text{tr}[\boldsymbol{V}])\psi(\boldsymbol{r}) \right\|^2/n \approx \|\boldsymbol{\varepsilon}\|^2/n + \|\boldsymbol{\Sigma}^{1/2}(\widehat{\boldsymbol{\beta}} - \boldsymbol{\beta}^*)\|^2. \quad (3)$$

48 Figure 1 illustrates the accuracy of (3) on simulated data. To study the criterion (2) and derive the
49 approximation (3), we develop novel results of independent interest regarding $M$-estimators in (1):

50 2. The paper derives general formula for the derivatives $(\partial/\partial y_i)\widehat{\boldsymbol{\beta}}$ and $(\partial/\partial x_{ij})\widehat{\boldsymbol{\beta}}$. This sheds light
51 on the differentiability structure of $M$-estimators for general loss-penalty pairs: for any $\rho, g$ with $g$
52 strongly convex, there exists $\widehat{\boldsymbol{A}} \in \mathbb{R}^{p \times p}$ depending on $(\boldsymbol{y}, \boldsymbol{X})$ such that for almost every $(\boldsymbol{y}, \boldsymbol{X})$,

$$(\partial/\partial y_i)\widehat{\boldsymbol{\beta}}(\boldsymbol{y}, \boldsymbol{X}) = \widehat{\boldsymbol{A}} \boldsymbol{X}^\top \boldsymbol{e}_i \psi'(r_i), \quad (\partial/\partial x_{ij})\widehat{\boldsymbol{\beta}}(\boldsymbol{y}, \boldsymbol{X}) = \widehat{\boldsymbol{A}} \boldsymbol{e}_j \psi(r_i) - \widehat{\boldsymbol{A}} \boldsymbol{X}^\top \boldsymbol{e}_i \psi'(r_i)\hat{\beta}_j,$$

53 for $r_i = y_i - \boldsymbol{x}_i^\top \hat{\boldsymbol{\beta}}, \forall i \in [n], j \in [p]$ where $\boldsymbol{e}_j \in \mathbb{R}^p$ and $\boldsymbol{e}_i \in \mathbb{R}^n$ are canonical basis vectors.

54 3. The paper obtains a stochastic representation for the residual $y_i - \boldsymbol{x}_i^\top \widehat{\boldsymbol{\beta}}$ for some fixed $i = 1, ..., n$,
55 extending some results of [10] on unregularized $M$-estimators to penalized ones as in (1). In
56 short, for each $i = 1, ..., n$ the $i$-th residual satisfies $r_i = y_i - \boldsymbol{x}_i^\top \widehat{\boldsymbol{\beta}}$

$$r_i + (\hat{\mathsf{df}}/\text{tr}\,\boldsymbol{V})\psi(r_i) \approx \varepsilon_i + Z_i \|\boldsymbol{\Sigma}^{1/2}(\widehat{\boldsymbol{\beta}} - \boldsymbol{\beta}^*)\| \quad (4)$$

57 where $Z_i \sim N(0, 1)$ is independent of $\varepsilon_i$. This stochastic representation is the motivation for
58 the criterion (2) as the amplitude of the normal part in the right-hand side is proportional to the
59 out-of-sample error $\|\boldsymbol{\Sigma}^{1/2}(\widehat{\boldsymbol{\beta}} - \boldsymbol{\beta}^*)\|$ that we wish to minimize, while the variance of the noise
60 $\varepsilon_i$ does not depend on the choice of $(\rho, g)$.

Simulated data in Figure 2 confirms that the stochastic representation for the $i$-th residual $r_i = y_i - \boldsymbol{x}_i^\top \widehat{\boldsymbol{\beta}}$ is accurate. Our working assumption throughout the paper is the following.

**Assumption 1.1.** *For constants $\gamma, \mu > 0$ independent of $n, p$ we have $p/n \leq \gamma$, the loss $\rho : \mathbb{R} \to \mathbb{R}$ is convex with a unique minimizer at 0, continuously differentiable and its derivative $\psi = \rho'$ is 1-Lipschitz. The design matrix $\boldsymbol{X}$ has iid $N(\boldsymbol{0}, \boldsymbol{\Sigma})$ rows for some invertible covariance $\boldsymbol{\Sigma}$ and the noise $\boldsymbol{\varepsilon}$ is independent of $\boldsymbol{X}$ with continuous distribution. The penalty $g : \mathbb{R}^p \to \mathbb{R}$ is $\mu$-strongly convex w.r.t. $\boldsymbol{\Sigma}$ in the sense that $\boldsymbol{b} \mapsto g(\boldsymbol{b}) - (\mu/2)\boldsymbol{b}^\top \boldsymbol{\Sigma} \boldsymbol{b}$ is convex in $\boldsymbol{b} \in \mathbb{R}^p$.*

Throughout the paper, we consider a sequence (say, indexed by $n$) of regression problems with $p$, $\boldsymbol{\beta}^*$, $\boldsymbol{\Sigma}$ and the loss-penalty pair $(\rho, g)$ depending implicitly on $n$. For some deterministic sequence $(a_n)$, the stochastically bounded notation $O_P(a_n)$ in this context may hide constants depending on $\gamma, \mu$ only, that is, $O_P(a_n)$ denotes a sequence of random variables $W_n$ such that for any $\varepsilon > 0$ there exists $K$ depending on $(\varepsilon, \gamma, \mu)$ satisfying $\mathbb{P}(|W_n| \geq K a_n) \leq \varepsilon$.

Since Assumption 1.1 requires $p/n \leq \gamma$, the Bolzano-Weierstrass theorem lets us extract a subsequence of regression problems such that $p/n \to \gamma'$ along this subsequence, for some constant $\gamma$. This is the asymptotic regime we have in mind throughout the paper, although our results do not require a specific limit for the ratio $p/n$. For some results, we will require the following additional assumption which is satisfied by robust loss functions and penalty that shrink towards 0.

**Assumption 1.2.** *The penalty is minimized at $\boldsymbol{0}$, that is, $g(\boldsymbol{0}) = \min_{\boldsymbol{b} \in \mathbb{R}^p} g(\boldsymbol{b})$; the loss is Lipschitz as in $|\psi| \leq M$ for some constant $M$ independent of $n, p$; the signal is bounded as in $\|\boldsymbol{\Sigma}^{1/2}\boldsymbol{\beta}^*\|^2 \leq M$.*

## 1.2 Related works

The context of the present work is the study of $M$-estimators in the regime $\frac{p}{n}$ has a finite limit. This literature pioneered in [2, 10, 9, 15] typically describes the subtle behavior of $\widehat{\boldsymbol{\beta}}$ in this regime by solving a system of nonlinear equations. This system typically depends on a prior distribution for the components of $\boldsymbol{\beta}^*$, and either depends on the covariance $\boldsymbol{\Sigma}$ [7] or assume $\boldsymbol{\Sigma} = \boldsymbol{I}_p$ [2, 16, 6, among many others]. Solutions to the nonlinear system are a powerful tool to understand $\widehat{\boldsymbol{\beta}}$ in theory, e.g., to characterize the deterministic limit of $\|\boldsymbol{\Sigma}^{1/2}(\widehat{\boldsymbol{\beta}} - \boldsymbol{\beta}^*)\|$, see e.g., the general results in [6] for the square loss and [16] for general loss-penalty pairs. However, since the system and its solution depend on unobservable quantity ($\boldsymbol{\Sigma}$ and prior on $\boldsymbol{\beta}^*$), the system solution is not directly usable for practical purposes such as parameter tuning.

The present work distinguishes itself from most of this literature as the goal is to describe the behavior of $\widehat{\boldsymbol{\beta}}$ using observable quantities that only depend on the data $(\boldsymbol{y}, \boldsymbol{X})$ (and not unobservable ones such as $\boldsymbol{\Sigma}$ or a prior distribution on $\boldsymbol{\beta}^*$ that appear in the aforementioned nonlinear system of equations). As we will see this view lets us perform adaptive tuning of parameters in a fully adaptive manner using the criterion (2). The criterion (2) appeared in previous works for the square loss only: [1, 12] studied (2) for the Lasso with $\boldsymbol{\Sigma} = \boldsymbol{I}_p$ and [3, Section 3] for the square loss and general penalty (note that for the square loss $\rho(u) = u^2/2$, (2) reduces to $n^2\|\boldsymbol{r}\|^2/(n - \hat{\mathsf{df}})^2$ due to $\psi(u) = u$ and $\mathrm{tr}[\boldsymbol{V}] = n - \hat{\mathsf{df}}$. The property $\psi(u) = u$ of the square loss hides the subtle interplay between $\boldsymbol{r}, \psi(\boldsymbol{r}), \hat{\mathsf{df}}$ and $\mathrm{tr}[\boldsymbol{V}]$ in (2) for $\rho$ different than the square loss). A criterion different from (2) is studied in [12, 3] to estimate the out-of-sample error. That criterion has the drawback to require the knowledge of $\boldsymbol{\Sigma}$, unlike (2) which is fully adaptive.

This work leverages probabilistic results on functions of standard normal random variables [4][3, §6, §7] which are consequences of Stein's formula [14]. Consequently, the main limitation of our work is that it currently requires Gaussian design for the probabilistic results (on the other hand, the differentiability result (5) is deterministic and does not rely on any probabilistic assumption).

## 2 Differentiability of regularized M-estimators

The first step towards the study of the criterion (2) is to justify the almost sure existence of the derivatives of $\widehat{\boldsymbol{\beta}}$ that appear in (2) through the scalar scalar $\hat{\mathsf{df}}$ and the matrix $\boldsymbol{V}$ in (2). Although the criterion (2) only involves the derivatives of $\widehat{\boldsymbol{\beta}}(\boldsymbol{y}, \boldsymbol{X})$ with respect to $\boldsymbol{y}$ for a fixed $\boldsymbol{X}$, the proof of

109 our results rely on the interplay between the derivatives with respect to $\boldsymbol{y}$ and with respect to $\boldsymbol{X}$: this
110 *differentiability structure* of $M$-estimators is the content of the following result.

111 **Theorem 2.1.** *Let Assumption 1.1 be fulfilled. For almost every* $(\boldsymbol{y}, \boldsymbol{X})$ *the map* $(\boldsymbol{y}, \boldsymbol{X}) \mapsto \widehat{\boldsymbol{\beta}}(\boldsymbol{y}, \boldsymbol{X})$
112 *is differentiable at* $(\boldsymbol{y}, \boldsymbol{X})$ *and there exists a matrix* $\widehat{\boldsymbol{A}} \in \mathbb{R}^{p \times p}$ *with* $\|\boldsymbol{\Sigma}^{1/2}\widehat{\boldsymbol{A}}\boldsymbol{\Sigma}^{1/2}\|_{op} \leq (n\mu)^{-1}$ *s.t.*

$$(\partial/\partial y_i)\widehat{\boldsymbol{\beta}}(\boldsymbol{y}, \boldsymbol{X}) = \widehat{\boldsymbol{A}}\boldsymbol{X}^\top \boldsymbol{e}_i \psi'(r_i),$$
$$(\partial/\partial x_{ij})\widehat{\boldsymbol{\beta}}(\boldsymbol{y}, \boldsymbol{X}) = \widehat{\boldsymbol{A}}\boldsymbol{e}_j \psi(r_i) - \widehat{\boldsymbol{A}}\boldsymbol{X}^\top \boldsymbol{e}_i \psi'(r_i)\hat{\beta}_j, \qquad where \ r_i = y_i - \boldsymbol{x}_i^\top \hat{\boldsymbol{\beta}}, \qquad (5)$$

113 $\boldsymbol{e}_i \in \mathbb{R}^n, \boldsymbol{e}_j \in \mathbb{R}^p$ *are canonical basis vectors ,* $\psi := \rho'$ *and* $\psi'$ *denote the derivatives. Furthermore,*

$$\hat{\mathsf{df}} = \mathrm{tr}[\boldsymbol{X}(\partial/\partial \boldsymbol{y})\hat{\boldsymbol{\beta}}] = \mathrm{tr}[\boldsymbol{X}\widehat{\boldsymbol{A}}\boldsymbol{X}\,\mathrm{diag}\{\psi'(\boldsymbol{r})\}], \qquad (6)$$

$$\boldsymbol{V} = \mathrm{diag}\{\psi'(\boldsymbol{r})\}(\boldsymbol{I}_n - \boldsymbol{X}(\partial/\partial \boldsymbol{y})\hat{\boldsymbol{\beta}}) = \mathrm{diag}\{\psi'(\boldsymbol{r})\} - \mathrm{diag}\{\psi'(\boldsymbol{r})\}\boldsymbol{X}\widehat{\boldsymbol{A}}\boldsymbol{X}\,\mathrm{diag}\{\psi'(\boldsymbol{r})\}. \quad (7)$$

114 *satisfy* $0 \leq \hat{\mathsf{df}} \leq n$ *and* $0 \leq \mathrm{tr}[\boldsymbol{V}] \leq n$.

115 Since the same matrix $\widehat{\boldsymbol{A}}$ appears in both the derivatives with respect to $y_i$ and to $x_{ij}$, (5) provides
116 relationship between $(\partial/\partial y_i)\hat{\boldsymbol{\beta}}$ and $(\partial/\partial x_{ij})\hat{\boldsymbol{\beta}}$, for instance $(\partial/\partial x_{ij})\hat{\boldsymbol{\beta}} = \widehat{\boldsymbol{A}}\boldsymbol{e}_j \psi(r_i) - \hat{\beta}_j(\partial/\partial y_i)\hat{\boldsymbol{\beta}}$.
117 Although the matrix $\widehat{\boldsymbol{A}}$ is not explicit for arbitrary loss-penalty pair, closed-form expressions are
118 available for particular examples such as the Elastic-Net penalty as discussed in Section 6.

119 **Remark 2.1.** *For the square loss* $\rho(u) = u^2/2$*, the differentiability formulae* (5) *reduce to*

$$(\partial/\partial y_l)\hat{\boldsymbol{\beta}}(\boldsymbol{y}, \boldsymbol{X}) = \widehat{\boldsymbol{A}}\boldsymbol{X}^\top \boldsymbol{e}_l, \qquad (\partial/\partial x_{ij})\hat{\boldsymbol{\beta}}(\boldsymbol{y}, \boldsymbol{X}) = \widehat{\boldsymbol{A}}\boldsymbol{e}_j(y_i - \boldsymbol{x}_i^\top \hat{\boldsymbol{\beta}}) - \widehat{\boldsymbol{A}}\boldsymbol{X}^\top \boldsymbol{e}_i \hat{\beta}_j$$

120 *for most every* $(\boldsymbol{y}, \boldsymbol{X})$ *and some matrix* $\widehat{\boldsymbol{A}} \in \mathbb{R}^{p \times p}$ *depending on* $(\boldsymbol{y}, \boldsymbol{X})$*, since in this case* $\psi' = 1$.

121 In the simple case where $g$ is twice continuously differentiable, (5) follows [4] with

$$\widehat{\boldsymbol{A}} = \left(\boldsymbol{X}^\top \mathrm{diag}\{\psi'(\boldsymbol{r})\}\boldsymbol{X} + n\nabla^2 g(\hat{\boldsymbol{\beta}})\right)^{-1} \qquad (8)$$

by differentiating the KKT conditions $\boldsymbol{X}^\top \psi(\boldsymbol{y} - \boldsymbol{X}\hat{\boldsymbol{\beta}}) = n\nabla g(\hat{\boldsymbol{\beta}})$. To illustrate why this is true,
provided that $\hat{\boldsymbol{\beta}}(\boldsymbol{y}, \boldsymbol{X})$ is differentiable, if $(\boldsymbol{y}(t), \boldsymbol{X}(t))$ are smooth perturbations of $(\boldsymbol{y}, \boldsymbol{X})$ with
$(\boldsymbol{y}(0), \boldsymbol{X}(0)) = (\boldsymbol{y}, \boldsymbol{X})$ and $\frac{d}{dt}(\boldsymbol{y}(t), \boldsymbol{X}(t))|_{t=0} = (\dot{\boldsymbol{y}}, \dot{\boldsymbol{X}})$, differentiation of $\boldsymbol{X}(t)^\top \psi(\boldsymbol{y}(t) - \boldsymbol{X}(t)\hat{\boldsymbol{\beta}}(\boldsymbol{y}(t), \boldsymbol{X}(t))) = n\nabla g(\hat{\boldsymbol{\beta}}(\boldsymbol{y}(t), \boldsymbol{X}(t)))$ at $t = 0$ and the chain rule yields

$$\dot{\boldsymbol{X}}^\top \psi(\boldsymbol{r}) - \boldsymbol{X}^\top \mathrm{diag}\{\psi'(\boldsymbol{r})\}(\dot{\boldsymbol{y}} - \dot{\boldsymbol{X}}\hat{\boldsymbol{\beta}}(\boldsymbol{y}, \boldsymbol{X})) = \widehat{\boldsymbol{A}}^{-1}\tfrac{d}{dt}\hat{\boldsymbol{\beta}}(\boldsymbol{y}(t), \boldsymbol{X}(t))\big|_{t=0}$$

122 with $\widehat{\boldsymbol{A}}$ in (8). This gives (5) if the penalty $g$ is twice-differentiable. Theorem 2.1 reveals that for
123 *arbitrary* convex penalty functions including non-differentiable ones, the differentiability structure
124 (5) always holds, as in the case of twice differentiable penalty $g$, even for penalty functions such as
125 $g(\boldsymbol{b}) = \mu\|\boldsymbol{b}\|^2/2 + \lambda\|\mathrm{mat}(\boldsymbol{b})\|_{\mathrm{nuc}}$ where $\mathrm{mat} : \mathbb{R}^p \to \mathbb{R}^{d_1 \times d_2}$ is a linear isomorphism to the space of
126 $d_1 \times d_2$ matrices and $\|\cdot\|_{\mathrm{nuc}}$ is the nuclear norm: in this case by Theorem 2.1 there exists a matrix
127 $\widehat{\boldsymbol{A}} \in \mathbb{R}^{p \times p}$ such that (5) holds although no closed-form expression for $\widehat{\boldsymbol{A}}$ is known.

128 The representation (5) is a powerful tool as it provides explicit derivatives of quantities of interest
129 such as $\boldsymbol{r} = \boldsymbol{y} - \boldsymbol{X}\hat{\boldsymbol{\beta}}$, $\|\psi(\boldsymbol{r})\|^2$ or $\|\boldsymbol{\Sigma}^{1/2}(\hat{\boldsymbol{\beta}} - \boldsymbol{\beta}^*)\|^2$. These explicit derivatives can then be used in
130 probabilistic identities and inequalities that involve derivatives, for instance Stein's formulae [14],
131 the Gaussian Poincaré inequalty [5, Theorem 3.20], or normal approximations [8, 4].

132 **Remark 2.2.** *Similar derivative formulae hold if an intercept is included in the minimization, as in*

$$\left(\hat{\beta}_0(\boldsymbol{y}, \boldsymbol{X}),\ \hat{\boldsymbol{\beta}}(\boldsymbol{y}, \boldsymbol{X})\right) = \operatorname*{argmin}_{b_0 \in \mathbb{R}, \boldsymbol{b} \in \mathbb{R}^p} \frac{1}{n}\sum_{i=1}^n \rho(y_i - b_0 - \boldsymbol{x}_i^\top \boldsymbol{b}) + g(\boldsymbol{b}) \qquad (9)$$

133 *Let Assumption 1.1 be fulfilled, and assume further* $\|\psi'(\boldsymbol{r})\|_2 > 0$ *with* $\boldsymbol{r} := \boldsymbol{y} - \boldsymbol{1}_n\hat{\beta}_0 - \boldsymbol{x}_i^\top \hat{\boldsymbol{\beta}}$ *where*
134 $\boldsymbol{1}_n = (1, ..., 1)^\top \in \mathbb{R}^n$*. For almost every* $(\boldsymbol{y}, \boldsymbol{X})$ *the map* $(\boldsymbol{y}, \boldsymbol{X}) \mapsto \hat{\boldsymbol{\beta}}(\boldsymbol{y}, \boldsymbol{X})$ *is differentiable at*
135 $(\boldsymbol{y}, \boldsymbol{X})$*, and there exists* $\widehat{\boldsymbol{A}} \in \mathbb{R}^{p \times p}$ *depending on* $(\boldsymbol{y}, \boldsymbol{X})$ *with* $\|\boldsymbol{\Sigma}^{1/2}\widehat{\boldsymbol{A}}\boldsymbol{\Sigma}^{1/2}\|_{op} \leq (n\mu)^{-1}$ *such that*
136

$$(\partial/\partial y_i)\hat{\boldsymbol{\beta}}(\boldsymbol{y}, \boldsymbol{X}) = \widehat{\boldsymbol{A}}\boldsymbol{X}^\top \boldsymbol{\Psi}' \boldsymbol{e}_i, \quad (\partial/\partial x_{ij})\hat{\boldsymbol{\beta}}(\boldsymbol{y}, \boldsymbol{X}) = \widehat{\boldsymbol{A}}\boldsymbol{e}_j \psi(r_i) - \widehat{\boldsymbol{A}}\boldsymbol{X}^\top \boldsymbol{\Psi}' \boldsymbol{e}_i \hat{\beta}_j, \qquad (10)$$

137 *where* $\boldsymbol{e}_i \in \mathbb{R}^n, \boldsymbol{e}_j \in \mathbb{R}^p$ *are canonical basis vectors,* $\psi = \rho'$ *and* $\boldsymbol{\Psi}' := \mathrm{diag}\{\psi'(\boldsymbol{r})\} - $
138 $\psi'(\boldsymbol{r})\psi'(\boldsymbol{r})^\top / \sum_{i \in [n]} \psi'(r_i)$.

## 3 Distribution of individual residuals

We now turn to the distribution of a single residual $r_i = y_i - \boldsymbol{x}_i^\top \widehat{\boldsymbol{\beta}}$ for some fixed observation $i \in \{1, ..., n\}$ (for instance, fix $i = 1$). By leveraging the differentiability structure (5) and the normal approximation from [4], the following result provides a clear picture of the distribution of $r_i$.

**Theorem 3.1.** *Let Assumption 1.1 be fulfilled and let $\widehat{\boldsymbol{A}} \in \mathbb{R}^{p \times p}$ be given by Theorem 2.1. Then for every $i = 1, ..., n$ there exists $Z_i \sim N(0,1)$ such that*

$$\left| \left( r_i + \mathrm{tr}[\boldsymbol{\Sigma}\widehat{\boldsymbol{A}}]\psi(r_i) \right) - \left( \varepsilon_i + \|\boldsymbol{\Sigma}^{1/2}(\widehat{\boldsymbol{\beta}} - \boldsymbol{\beta}^*)\| Z_i \right) \right| \le O_P(n^{-1/4})(|\psi(\varepsilon_i)| + \|\boldsymbol{\Sigma}^{1/2}(\widehat{\boldsymbol{\beta}} - \boldsymbol{\beta}^*)\|) \quad (11)$$

*Furthermore, if $\varepsilon_i$ has a fixed distribution $F$, there exists a bivariate variable $(\tilde{\varepsilon}_i^n, \tilde{Z}_i^n)$ converging in distribution to the product measure $F \otimes N(0,1)$ such that*

$$r_i + \mathrm{tr}[\boldsymbol{\Sigma}\widehat{\boldsymbol{A}}]\psi(r_i) = \tilde{\varepsilon}_i^n + \|\boldsymbol{\Sigma}^{1/2}(\widehat{\boldsymbol{\beta}} - \boldsymbol{\beta}^*)\|\tilde{Z}_i^n. \quad (12)$$

*If $\varepsilon_i$ has a fixed distribution $F$ and Assumption 1.2 holds then $|\psi(\varepsilon_i)| + \|\boldsymbol{\Sigma}^{1/2}(\widehat{\boldsymbol{\beta}} - \boldsymbol{\beta}^*)\| = O_P(1)$.*

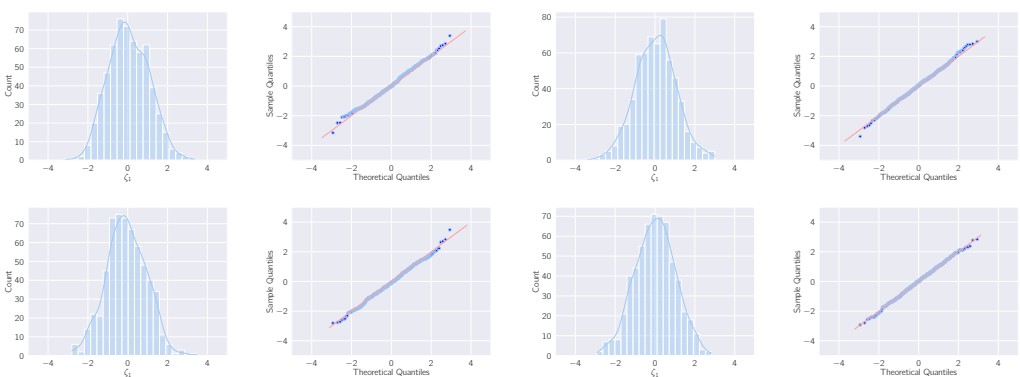

Figure 2: Histogram and QQ-plot for $\zeta_1$ in (13) under Huber Elastic-Net regression for different choices of tuning parameters $(\lambda, \tau)$. Left Top: $(0.036, 10^{-10})$, Right Top: $(0.054, 0.01)$, Left Bottom: $(0.036, 0.01)$, Right Bottom: $(0.024, 0.1)$. Each figure contains 600 data points generated with anisotropic design matrix and iid $\varepsilon_i$ from the $t$-distribution with 2 degrees of freedom. A detailed setup is provided in Section 6.

Theorem 3.1 is a formal statement regarding the informal normal approximation

$$\zeta_i := \frac{r_i + \mathrm{tr}[\boldsymbol{\Sigma}\widehat{\boldsymbol{A}}]\psi(r_i) - \varepsilon_i}{\|\boldsymbol{\Sigma}^{1/2}(\hat{\boldsymbol{\beta}} - \boldsymbol{\beta}^*)\|} \approx N(0,1). \quad (13)$$

Simulations in Figure 2 confirm the normality of $\zeta_i$ for the Huber loss with Elastic-Net penalty and four combinations of tuning parameters. For the square loss $\rho(u) = u^2/2$, because $\psi(u) = u$, asymptotic normality of the residuals hold in the following form.

**Theorem 3.2.** *Let Assumption 1.1 hold with $\rho(u) = u^2/2$ and $\boldsymbol{\varepsilon} \sim N(\boldsymbol{0}, \sigma^2 \boldsymbol{I}_n)$. Then for $i = 1$,*

$$(\sigma^2 + \|\boldsymbol{\Sigma}^{1/2}(\widehat{\boldsymbol{\beta}} - \boldsymbol{\beta}^*)\|^2)^{-1/2}(1 + \mathrm{tr}[\boldsymbol{\Sigma}\widehat{\boldsymbol{A}}])(y_i - \boldsymbol{x}_i^\top \widehat{\boldsymbol{\beta}}) \to^d N(0,1) \qquad as\ n \to +\infty. \quad (14)$$

It is informative to provide a sketch of the proof of Theorem 3.1 explain the appearance of $\psi(r_i)$ and $\mathrm{tr}[\boldsymbol{\Sigma}\widehat{\boldsymbol{A}}]$ in the normal approximation results (11) and (13). A variant of the normal approximation of [4] proved in the supplement states that for a differentiable function $\mathbf{f} : \mathbb{R}^q \to \mathbb{R}^q \setminus \{\boldsymbol{0}\}$ and $\boldsymbol{z} \sim N(\boldsymbol{0}, \boldsymbol{I}_q)$, there exists $Z \sim N(0,1)$ such hat

$$\mathbb{E}\left[\left| \frac{\mathbf{f}(\boldsymbol{z})^\top \boldsymbol{z} - \sum_{k=1}^q (\partial/\partial z_k) f_k(\boldsymbol{z})}{\|\mathbf{f}(\boldsymbol{z})\|} - Z \right|^2\right] \le C_1 \mathbb{E}\left[ \frac{\sum_{k=1}^q \|(\partial/\partial z_k)\mathbf{f}(\boldsymbol{z})\|^2}{\|\mathbf{f}(\boldsymbol{z})\|^2} \right]. \quad (15)$$

Some technical hurdles aside, the proof sketch is the following: Apply the previous display to $q = p$, $\boldsymbol{z} = \boldsymbol{\Sigma}^{-1/2}\boldsymbol{x}_i$ conditionally on $(\boldsymbol{\varepsilon}, (\boldsymbol{x}_l)_{l \in [n] \setminus \{i\}})$ and to $\mathbf{f}(\boldsymbol{z}) = \boldsymbol{\Sigma}^{1/2}(\widehat{\boldsymbol{\beta}} - \boldsymbol{\beta}^*)$ in the simple case where $\boldsymbol{\beta}^* = 0$ (this amounts to performing a change of variable by translation of $\widehat{\boldsymbol{\beta}}$ to $\widehat{\boldsymbol{\beta}} - \boldsymbol{\beta}^*$). Then the right-hand side of the previous display is negligible in probability compared to $Z$, and in the left-hand side $\mathbf{f}(\boldsymbol{z})^\top \boldsymbol{z} = \boldsymbol{x}_i^\top(\widehat{\boldsymbol{\beta}} - \boldsymbol{\beta}^*)$ and $\sum_{k=1}^q (\partial/\partial z_k) f_k(\boldsymbol{z}) \approx \mathrm{tr}[\boldsymbol{\Sigma}\widehat{\boldsymbol{A}}]\psi(r_i)$ as the second term in (5) is negligible. This completes the sketch of the proof of (13).

**Proximal operator representation.** From the above asymptotic normality results, a stochastic representation for the $i$-th residual $r_i = y_i - \boldsymbol{x}_i^\top \widehat{\boldsymbol{\beta}}$ can be obtained as follows: With $\text{prox}[t\rho](u)$ the proximal operator of $x \mapsto t\rho(x)$ defined as the unique solution $z \in \mathbb{R}$ of equation $z + t\psi(z) = u$,

$$r_i = y_i - \boldsymbol{x}_i^\top \widehat{\boldsymbol{\beta}} = \text{prox}[\hat{t}\rho]\big(\tilde{\varepsilon}_i^n + \|\boldsymbol{\Sigma}^{1/2}(\widehat{\boldsymbol{\beta}} - \boldsymbol{\beta}^*)\|\tilde{Z}_i^n\big) \qquad \text{with } \hat{t} = \text{tr}[\boldsymbol{\Sigma}\widehat{\boldsymbol{A}}]$$

where $(\tilde{\varepsilon}_i^n, \tilde{Z}_i^n)$ converges in distribution to product measure $F \otimes N(0,1)$ where $F$ is the law of $\varepsilon_i$.

# 4 A proxy of the out-of-sample error if $\boldsymbol{\Sigma}$ is known

The approximations of the previous sections for $r_i + \text{tr}[\boldsymbol{\Sigma}\widehat{\boldsymbol{A}}]\psi(r_i)$ and the fact that $\varepsilon_i$ is independent of $Z_i \sim N(0,1)$ in (11) suggest that $(r_i + \text{tr}[\boldsymbol{\Sigma}\widehat{\boldsymbol{A}}]\psi(r_i))^2 \approx \varepsilon_i^2 + \|\boldsymbol{\Sigma}^{1/2}(\widehat{\boldsymbol{\beta}} - \boldsymbol{\beta}^*)\|^2 Z_i^2$; and averaging over $\{1, ..., n\}$ one can hope for the approximation $\|\boldsymbol{r} + \text{tr}[\boldsymbol{\Sigma}\widehat{\boldsymbol{A}}]\psi(\boldsymbol{r})\|^2/n \approx \|\boldsymbol{\varepsilon}\|^2/n + \|\boldsymbol{\Sigma}^{1/2}(\widehat{\boldsymbol{\beta}} - \boldsymbol{\beta}^*)\|^2$. The following result makes this heuristic precise.

**Theorem 4.1.** *Let Assumption 1.1 be fulfilled and $\widehat{\boldsymbol{A}}$ be given by Theorem 2.1. Then*

$$\|\boldsymbol{\Sigma}^{1/2}(\widehat{\boldsymbol{\beta}} - \boldsymbol{\beta}^*)\|^2 + \|\boldsymbol{\varepsilon}\|^2/n = \big\|\boldsymbol{r} + \text{tr}[\boldsymbol{\Sigma}\widehat{\boldsymbol{A}}]\psi(\boldsymbol{r})\big\|^2/n + O_P(n^{-1/2})\,\text{Rem},$$

*where* $\text{Rem} := \|\boldsymbol{\Sigma}^{1/2}(\hat{\boldsymbol{\beta}} - \boldsymbol{\beta}^*)\|^2 + \frac{1}{n}\|\psi(\boldsymbol{r})\|^2 + (\|\boldsymbol{\Sigma}^{1/2}(\hat{\boldsymbol{\beta}} - \boldsymbol{\beta}^*)\|^2 + \frac{1}{n}\|\psi(\boldsymbol{r})\|^2)^{1/2}\|\frac{1}{\sqrt{n}}\boldsymbol{\varepsilon}\|$. *Thus*

$$\|\boldsymbol{\Sigma}^{1/2}(\widehat{\boldsymbol{\beta}} - \boldsymbol{\beta}^*)\|^2 + \|\boldsymbol{\varepsilon}\|^2/n = (1 + O_P(n^{-1/2}))\big\|\boldsymbol{r} + \text{tr}[\boldsymbol{\Sigma}\widehat{\boldsymbol{A}}]\psi(\boldsymbol{r})\big\|^2/n.$$

Theorem 4.1 provides a first candidate, $\big\|\boldsymbol{r} + \text{tr}[\boldsymbol{\Sigma}\widehat{\boldsymbol{A}}]\psi(\boldsymbol{r})\big\|^2/n$ to estimate

$$\|\boldsymbol{\Sigma}^{1/2}(\widehat{\boldsymbol{\beta}} - \boldsymbol{\beta}^*)\|^2 + \|\boldsymbol{\varepsilon}\|^2/n. \tag{16}$$

Estimation of (16) is useful as $\|\boldsymbol{\varepsilon}\|^2/n$ is independent of the choice of the estimator $\widehat{\boldsymbol{\beta}}$ and in particular independent of the chosen loss-penalty pair in (1). Given two or more estimators (1), choosing the one with smallest $\big\|\boldsymbol{r} + \text{tr}[\boldsymbol{\Sigma}\widehat{\boldsymbol{A}}]\psi(\boldsymbol{r})\big\|^2$ is thus a good proxy for minimizing the out-of-sample error.

**Corollary 4.2.** *Let $\widehat{\boldsymbol{\beta}}, \widetilde{\boldsymbol{\beta}}$ be two M-estimators (1) Assumption 1.1 with loss-penalty pair $(\rho, g)$ and $(\tilde{\rho}, \tilde{g})$ respectively. Assume that both satisfy Assumption 1.1 and let $\psi = \rho'$ and $\tilde{\psi} = \tilde{\rho}'$. Let $\boldsymbol{r} = \boldsymbol{y} - \boldsymbol{X}\widehat{\boldsymbol{\beta}}, \widetilde{\boldsymbol{r}} = \boldsymbol{y} - \boldsymbol{X}\widetilde{\boldsymbol{\beta}}$ be the residuals, $\widehat{\boldsymbol{A}}, \widetilde{\boldsymbol{A}}$ be the corresponding matrices of size $p \times p$ given by Theorem 2.1. Further assume that both estimators satisfy Assumption 1.2 and that $\boldsymbol{\varepsilon}$ has iid coordinates independent with $\mathbb{E}[|\varepsilon_i|^{1+q}] \leq M$ for constants $q \in (0, 1), M > 0$ independent of $n, p$. Let $\Omega = \{\|\boldsymbol{X}\boldsymbol{\Sigma}^{-1/2}\|_{op} \leq 2\sqrt{n} + \sqrt{p}\} \cap \{\|\boldsymbol{\varepsilon}\|^2 \leq n^{2/(1+q)}\}$. Then for any $\eta > 0$ independent of $n, p$ there exists $C(\gamma, \mu, \eta, q, M) > 0$ depending only on $\{\gamma, \mu, \eta, q, M\}$ such that*

$$\mathbb{P}\Big(\|\boldsymbol{\Sigma}^{1/2}(\widehat{\boldsymbol{\beta}} - \boldsymbol{\beta}^*)\|^2 - \|\boldsymbol{\Sigma}^{1/2}(\widetilde{\boldsymbol{\beta}} - \boldsymbol{\beta}^*)\|^2 > \eta, \;\; \|\boldsymbol{r} + \text{tr}[\boldsymbol{\Sigma}\widehat{\boldsymbol{A}}]\psi(\boldsymbol{r})\|^2 \leq \|\widetilde{\boldsymbol{r}} + \text{tr}[\boldsymbol{\Sigma}\widetilde{\boldsymbol{A}}]\tilde{\psi}(\widetilde{\boldsymbol{r}})\|^2\Big)$$
$$\leq C(\gamma, \mu, \eta, q, M)n^{-q/(1+q)} + \mathbb{P}(\Omega^c) \to 0.$$

Provided that the noise random variables $\varepsilon_i$ have at least $1 + q$ moments, Corollary 4.2 implies that with probability approaching one given two M-estimators $\widehat{\boldsymbol{\beta}}$ and $\widetilde{\boldsymbol{\beta}}$, choosing the estimator corresponding to the smallest criteria among $\|\boldsymbol{r} + \text{tr}[\boldsymbol{\Sigma}\widehat{\boldsymbol{A}}]\boldsymbol{r}\|^2$ and $\|\widetilde{\boldsymbol{r}} + \text{tr}[\boldsymbol{\Sigma}\widetilde{\boldsymbol{A}}]\widetilde{\boldsymbol{r}}\|^2$ leads to the smallest out-of-sample error, up to any small constant $\eta > 0$. This allows noise random variables $\varepsilon_i$ with infinite variance. A similar result can be obtained to select among $K$ different M-estimators (1).

**Corollary 4.3.** *As in Corollary 4.2, assume $\mathbb{E}[|\varepsilon_i|^{1+q}] \leq M$ and let $\widehat{\boldsymbol{\beta}}_1, ..., \widehat{\boldsymbol{\beta}}_K$ be M-estimators of the form (1) with loss-penalty pair $(\rho_k, g_k)$ satisfying Assumptions 1.1 and 1.2. For each $k = 1, ..., K$, let $\boldsymbol{r}_k = \boldsymbol{y} - \boldsymbol{X}\widehat{\boldsymbol{\beta}}_k$ be the residuals and $\widehat{\boldsymbol{A}}_k$ be the corresponding matrix of size $p \times p$ from Theorem 2.1. Let $\hat{k} \in \arg\min_{k=1,...,K} \|\boldsymbol{r}_k + \text{tr}[\boldsymbol{\Sigma}\widehat{\boldsymbol{A}}_k]\psi_k(\boldsymbol{r}_k)\|$ where $\psi_k = \rho_k'$. Then if $(\gamma, \mu, \eta, q, M)$ are constants independent of $n, p$*

$$\mathbb{P}\big(\|\boldsymbol{\Sigma}^{1/2}(\widehat{\boldsymbol{\beta}}_{\hat{k}} - \boldsymbol{\beta}^*)\|^2 > \min_{k=1,...,K}\|\boldsymbol{\Sigma}^{1/2}(\widehat{\boldsymbol{\beta}}_k - \boldsymbol{\beta}^*)\|^2 + \eta\big) \to 0 \qquad \text{if } K = o(n^{q/(1+q)}).$$

191 Given $K$ different loss-penalty pairs and the corresponding $M$-estimators in (1), minimizing the
192 criterion $\|\boldsymbol{r} + \mathrm{tr}[\boldsymbol{\Sigma}\widehat{\boldsymbol{A}}]\boldsymbol{r}\|$ thus provably selects a loss-penalty pair that leads to an optimal out-
193 of-sample error, up to an arbitrary small constant $\eta > 0$ independent of $n, p$. The requirement
194 $K = o(n^{q/(1+q)})$ means that the cardinality of the collection of $M$-estimators to select from should
195 grow more slowly than a power of $n$. This is typically satisfied for default tuning parameter grids in
196 popular libraries (e.g., `sklearn.linear_model.Lasso` [13]) with tuning parameters evenly spaced
197 in a log-scale that consequently have cardinality logarithmic in the parameter range. The major
198 drawback of the criterion $\|\boldsymbol{r} + \mathrm{tr}[\boldsymbol{\Sigma}\widehat{\boldsymbol{A}}]\boldsymbol{r}\|$ is the dependence through $\mathrm{tr}[\boldsymbol{\Sigma}\widehat{\boldsymbol{A}}]$ on the covariance $\boldsymbol{\Sigma}$
199 of the design, which is typically unknown. The next section introduces an estimator of $\mathrm{tr}[\boldsymbol{\Sigma}\widehat{\boldsymbol{A}}]$ that
200 does not require the knowledge of $\boldsymbol{\Sigma}$.

## 5   Degrees of freedom and estimating $\mathrm{tr}[\boldsymbol{\Sigma}\widehat{\boldsymbol{A}}]$ without the knowledge of $\boldsymbol{\Sigma}$

This section focuses on estimating $\mathrm{tr}[\boldsymbol{\Sigma}\widehat{\boldsymbol{A}}]$. The matrix $\widehat{\boldsymbol{A}}$ from Theorem 2.1 can estimated from
the data $(\boldsymbol{y}, \boldsymbol{X})$ in the sense that $\widehat{\boldsymbol{A}}$ is a measurable function of $(\boldsymbol{y}, \boldsymbol{X})$ (thanks to the observation
that derivatives are limits, and limits of measurable functions are again measurable). The difficulty
is thus to estimate $\mathrm{tr}[\boldsymbol{\Sigma}\widehat{\boldsymbol{A}}]$ without the knowledge of $\boldsymbol{\Sigma}$. To illustrate this difficulty, consider
Ridge regression with square loss $\rho(u) = u^2/2$ and penalty $g(\boldsymbol{b}) = \tau\|\boldsymbol{b}\|^2/2$. Then $\widehat{\boldsymbol{\beta}}(\boldsymbol{y}, \boldsymbol{X}) = (\boldsymbol{X}^\top\boldsymbol{X} + \tau n\boldsymbol{I}_p)^{-1}\boldsymbol{X}^\top\boldsymbol{y}$ and $\widehat{\boldsymbol{A}}$ in Theorem 2.1 is given explicitly by $\widehat{\boldsymbol{A}} = (\boldsymbol{X}^\top\boldsymbol{X} + \tau n\boldsymbol{I}_p)^{-1}$ and

$$\mathrm{tr}[\boldsymbol{\Sigma}\widehat{\boldsymbol{A}}] = \mathrm{tr}[(\boldsymbol{G}^\top\boldsymbol{G} + n\tau\boldsymbol{\Sigma}^{-1})^{-1}], \qquad \text{where } \boldsymbol{G} = \boldsymbol{X}\boldsymbol{\Sigma}^{-1/2}.$$

202 Above, $\boldsymbol{G}$ is a random matrix with iid $N(0,1)$ entries the value of $\mathrm{tr}[\boldsymbol{\Sigma}\widehat{\boldsymbol{A}}]$ is highly dependent on the
203 spectrum of $\boldsymbol{\Sigma}^{-1}$. In this particular case, the limit of $\mathrm{tr}[(\boldsymbol{G}^\top\boldsymbol{G} + n\tau\boldsymbol{\Sigma}^{-1})^{-1}]$ can be obtained using
204 random matrix theory [11] as the limiting behavior of the Stieltjes transform of $\boldsymbol{G}^\top\boldsymbol{G}/n + \tau\boldsymbol{\Sigma}^{-1}$
205 and its spectral distribution is known; however the limit of the spetral distribution depends on the
206 spectrum of $\tau\boldsymbol{\Sigma}^{-1}$. This is not desirable here as we wish to construct estimators that require no
207 knowledge on $\boldsymbol{\Sigma}$. For more involved loss-penalty pairs such as the Elastic-Net in Example 6.1, such
208 random matrix theory results do not apply as $\mathrm{tr}[\boldsymbol{\Sigma}\widehat{\boldsymbol{A}}]$ depends on the random support of $\widehat{\boldsymbol{\beta}}$.

209 Instead, we do not rely on known random matrix theory results. With the matrix $\widehat{\boldsymbol{A}} \in \mathbb{R}^{p \times p}$ given by
210 Theorem 2.1, our proposal to estimate $\mathrm{tr}[\boldsymbol{\Sigma}\widehat{\boldsymbol{A}}]$ is the ratio $\widehat{\mathrm{df}}/\mathrm{tr}[\boldsymbol{V}]$ with $\widehat{\mathrm{df}}$ and $\boldsymbol{V}$ in (6)-(7). Both
211 the scalar $\widehat{\mathrm{df}}$ and the matrix $\boldsymbol{V} \in \mathbb{R}^{n \times n}$ are observable; in particular they do not depend on $\boldsymbol{\Sigma}$.

212 **Theorem 5.1.** *Let Assumption 1.1 be fulfilled and $\widehat{\boldsymbol{A}}$ be given by Theorem 2.1. Then*

$$\mathbb{E}[|\mathrm{tr}[\boldsymbol{\Sigma}\widehat{\boldsymbol{A}}]\,\mathrm{tr}[\boldsymbol{V}]/n - \widehat{\mathrm{df}}/n|] \leq C_2(\gamma, \mu)n^{-1/2}. \tag{17}$$

213 Simulations in Figure 3 and Table 1 confirm that the approximation $\mathrm{tr}[\boldsymbol{\Sigma}\widehat{\boldsymbol{A}}] \approx \widehat{\mathrm{df}}/\mathrm{tr}[\boldsymbol{V}]$ is accurate
214 for the Huber loss with Elastic-Net penalty. For the square loss, $\psi' = 1$ and $\mathrm{tr}[\boldsymbol{V}] = n - \widehat{\mathrm{df}}$ so that
215 (17) becomes $\mathbb{E}|(1 - \widehat{\mathrm{df}}/n)(1 + \mathrm{tr}[\boldsymbol{\Sigma}\widehat{\boldsymbol{A}}]) - 1| \leq C_3(\gamma, \mu)n^{-1/2}$ and the following result holds.
216 **Corollary 5.2.** *Let Assumption 1.1 be fulfilled with $\rho(u) = u^2/2$ and $\boldsymbol{\varepsilon} \sim N(\boldsymbol{0}, \sigma^2\boldsymbol{I}_n)$. Then*
217 $(1 - \widehat{\mathrm{df}}/n)(1 + \mathrm{tr}[\boldsymbol{\Sigma}\widehat{\boldsymbol{A}}]) \xrightarrow{\mathbb{P}} 1$ *and the normality* (14) *holds with* $1 + \mathrm{tr}[\boldsymbol{\Sigma}\widehat{\boldsymbol{A}}]$ *replaced by* $(1 - \widehat{\mathrm{df}}/n)^{-1}$.

218 For general loss $\rho$, the criterion (2) replaces $\mathrm{tr}[\boldsymbol{\Sigma}\widehat{\boldsymbol{A}}]$ by $\widehat{\mathrm{df}}/\mathrm{tr}[\boldsymbol{V}]$ in the proxy of the out-of-sample
219 error $\|\boldsymbol{r} + \mathrm{tr}[\boldsymbol{\Sigma}\widehat{\boldsymbol{A}}]\psi(\boldsymbol{r})\|^2$ studied in the previous section. Thanks to (17), this replacement preserves
220 the good properties of $\|\boldsymbol{r} + \mathrm{tr}[\boldsymbol{\Sigma}\widehat{\boldsymbol{A}}]\psi(\boldsymbol{r})\|^2$ proved in Corollaries 4.2 and 4.3.

**Theorem 5.3.** *For $k = 1, ..., K$, let $(\rho_k, g_k)$ be a loss-penalty pair satisfying Assumptions 1.1 and 1.2
with $\psi_k = \rho'_k$, let $\widehat{\boldsymbol{\beta}}_k, \boldsymbol{r}_k, \widehat{\boldsymbol{A}}_k$ be the corresponding $M$-estimator residual vector and matrix of size
$p \times p$ given by Theorem 2.1 as in Corollary 4.3 and let $\widehat{\mathrm{df}}_k = \mathrm{tr}[\boldsymbol{X}\boldsymbol{A}_k\boldsymbol{X}^\top \mathrm{diag}\{\psi'_k(\boldsymbol{r}_k)\}]$ and
$\boldsymbol{V}_k = \mathrm{diag}\{\psi'_k(\boldsymbol{r}_k)\}(\boldsymbol{I}_n - \boldsymbol{X}\boldsymbol{A}_k\boldsymbol{X}^\top \mathrm{diag}\{\psi'_k(\boldsymbol{r}_k)\})$. For a small constant $\eta > 0$ independent of
$n, p$, say $\eta = 0.05$, define*

$$\hat{k} \in \underset{k=1,...,K}{\mathrm{argmin}} \left\|\boldsymbol{r}_k + \frac{\widehat{\mathrm{df}}_k}{\mathrm{tr}[\boldsymbol{V}_k]}\psi_k(\boldsymbol{r}_k)\right\|^2 \qquad \textit{subject to} \qquad \frac{1}{n}\sum_{i=1}^{n}\psi'_k(r_{ki}) \geq \eta.$$

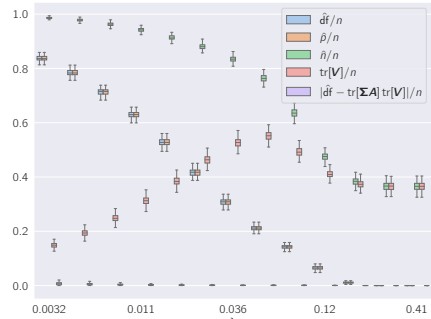 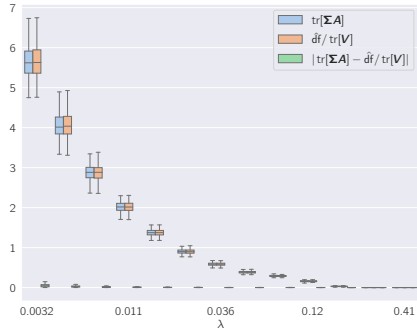

Figure 3: Above: Boxplots for $\hat{\mathrm{df}}, \hat{p}, \hat{n}, \mathrm{tr}[\boldsymbol{V}], \mathrm{tr}[\boldsymbol{\Sigma}\widehat{\boldsymbol{A}}]$ and $|\mathrm{tr}[\boldsymbol{\Sigma}\widehat{\boldsymbol{A}}] - \hat{\mathrm{df}}/\mathrm{tr}[\boldsymbol{V}]|$ in Huber Elastic-Net regression with $\tau = 10^{-10}$ and $\lambda \in [0.0032, 0.41]$. Each box contains 200 data points. Below: heatmaps for $\hat{\mathrm{df}}/n$, $\mathrm{tr}[\boldsymbol{V}]/n$ and $\hat{n}/n = \sum_{i=1}^{n} \psi'(r_i)/n$ under the simulation setup in Figure 1. The detailed simulation setup is given in Section 6.

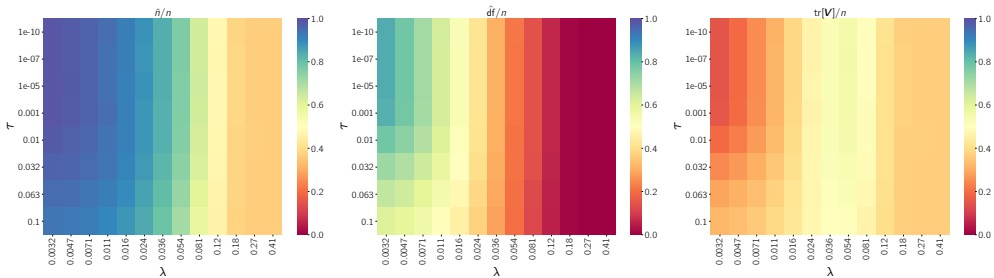

*If $\varepsilon_i$ has $1 + q$ moments in the sense that $\mathbb{E}[|\varepsilon_i|^{1+q}] \leq M$ for constants $q \in (0, 1), M > 0$. If $(M, q, \eta, \mu, \gamma)$ and $\tilde{\eta} > 0$ are independent of $n, p$ then*

$$\mathbb{P}\Big(\|\boldsymbol{\Sigma}^{1/2}(\widehat{\boldsymbol{\beta}}_{\hat{k}} - \boldsymbol{\beta}^*)\| > \min_{k=1,...,K:\frac{1}{n}\sum_{i=1}^{n}\psi'_k(r_{ki})\geq\eta} \|\boldsymbol{\Sigma}^{1/2}(\widehat{\boldsymbol{\beta}}_k - \boldsymbol{\beta}^*)\| + \tilde{\eta}\Big) \to 0 \qquad \text{if } K = o(n^{q/(1+q)}).$$

Figure 1 illustrates on simulations the success of the criterion (2) over a grid of tuning parameters for $M$-estimators with the Huber loss and Elastic-Net penalty. The criterion (2) is thus successful at selecting a $M$-estimator with smallest out-of-sample error up to an additive constant $\tilde{\eta}$, among those $M$-estimators indexed in $\{1, ..., K\}$ that are such that $\frac{1}{n}\sum_{i=1}^{n}\psi'_k(r_{ki}) \geq \eta$. On the one hand it is unclear to us whether the restriction $\frac{1}{n}\sum_{i=1}^{n}\psi'_k(r_{ki}) \geq \eta$; on the other hand there is a practical meaning in excluding $M$-estimators with small $\frac{1}{n}\sum_{i=1}^{n}\psi'_k(r_{ki})$: For the Huber loss $H(u) := u^2/2$ for $|u| \leq 1$ and $|u| - 1/2$ for $|u| \geq 1$ the quantity $\frac{1}{n}\sum_{i=1}^{n}\psi'_k(r_{ki})$ is the number of of data points in $\{1, ..., n\}$ such that the residual $y_i - \boldsymbol{x}_i^\top\widehat{\boldsymbol{\beta}}_k$ fall within the quadratic regime of the loss function. Observations $i \in \{1, ..., n\}$ that fall in the linear regime of the loss are excluded from the fit, in the sense that for some $i$ with $r_{ki} = y_i - \boldsymbol{x}_i^\top\widehat{\boldsymbol{\beta}}_k > 1$, replacing $y_i$ by $\tilde{y}_i = y_i + 1000$ (or any positive value) does not change the $M$-estimator solution $\widehat{\boldsymbol{\beta}}_k$ (this can be seen from the KKT conditions directly, or by integration the derivative with respect to $y_i$ in (5)). Thus the constraint $\frac{1}{n}\sum_{i=1}^{n}\psi'_k(r_{ki}) \geq \eta$ requires that at most a constant fraction of the observations are excluded from the fit (or equivalently, at least a constant fraction of the $n$ observations participate in the fit). For scaled versions of the Huber loss, $\rho_k(u) = a^2 H(a^{-1}u)$ for some $a > 0$, the value $\hat{n} = \frac{1}{n}\sum_{i=1}^{n}\psi'_k(r_{ki})$ again counts the number of residuals falling in the quadratic regime of the loss, i.e., the number of observations participating in the fit. The heatmaps of Figure 3 illustrate $\hat{n}$ in a simulation for a wide range of parameters. Similarly, for smooth robust loss functions such as $\rho_k(u) = \sqrt{1 + u^2}$, the constraint $\frac{1}{n}\sum_{i=1}^{n}\psi'_k(r_{ki}) \geq \eta$ requires that at most a constant fraction of the $n$ observations are such that $\psi'_k(r_{ki}) < \eta/2$, i.e., such that the second derivative $\psi'_k$ is too small (and the loss $\rho_k$ too flat).

Theorems 2.1, 3.2, 4.1 and 5.1 provide our general results applicable to a single regularized $M$-estimator (1) while corollaries such as Theorem 5.3 are obtained using the union bound. The next

245 section specializes our results and notation to the Huber loss with Elastic-Net penalty and details the
246 simulation setup used in the figures.

## 6   Example and simulation setting: Huber loss with Elastic-Net penalty

248 In simulations and in the example below, we focus on the loss-penalty pair

$$\rho(u;\Lambda) = \Lambda^2 H(\Lambda^{-1}u), \qquad g(\boldsymbol{b};\lambda,\tau) = \lambda\|\boldsymbol{b}\|_1 + (\tau/2)\|\boldsymbol{b}\|_2^2 \tag{18}$$

249 for tuning parameters $\Lambda,\lambda,\tau \geq 0$ where $H(u) := u^2/2$ for $|u| \leq 1$ and $|u| - 1/2$ for $|u| \geq 1$.

250 **Example 6.1.** *With $(\rho, g)$ in (18), matrix $\widehat{\boldsymbol{A}}$ in (5) matrix $\boldsymbol{V}$ in (7) and $\hat{\mathsf{df}}$ in (6) we have*

$$\widehat{\boldsymbol{A}}_{\hat{S},\hat{S}} = (\boldsymbol{X}_{\hat{S}}^\top \operatorname{diag}\{\psi'(\boldsymbol{r})\}\boldsymbol{X}_{\hat{S}} + n\tau\boldsymbol{I}_{\hat{p}})^{-1}, \quad A_{i,j} = 0 \text{ if } i \notin \hat{S} \text{ or } j \notin \hat{S},$$

$$\boldsymbol{V} = \operatorname{diag}\{\psi'(\boldsymbol{r})\} - \operatorname{diag}\{\psi'(\boldsymbol{r})\}\boldsymbol{X}_{\hat{S}}(\boldsymbol{X}_{\hat{S}}^\top \operatorname{diag}\{\psi'(\boldsymbol{r})\}\boldsymbol{X}_{\hat{S}} + n\tau\boldsymbol{I}_{\hat{p}})^{-1}\boldsymbol{X}_{\hat{S}}^\top \operatorname{diag}\{\psi'(\boldsymbol{r})\}, \tag{19}$$

$$\hat{\mathsf{df}} = \operatorname{tr}[\boldsymbol{X}_{\hat{S}}(\boldsymbol{X}_{\hat{S}}^\top \operatorname{diag}\{\psi'(\boldsymbol{r})\}\boldsymbol{X}_{\hat{S}} + n\tau\boldsymbol{I}_{\hat{p}})^{-1}\boldsymbol{X}_{\hat{S}}^\top \operatorname{diag}\{\psi'(\boldsymbol{r})\}],$$

251 *where $\hat{S}$ is the active set $\{j \in [p] : \hat{\beta}_j \neq 0\}$ and $\hat{p}$ is the size of $\hat{S}$; $\boldsymbol{X}_{\hat{S}}$ is the submatrix of $\boldsymbol{X}$*
252 *selecting columns with index in $\hat{S}$ and $\widehat{\boldsymbol{A}}_{\hat{S},\hat{S}}$ is the submatrix of $\widehat{\boldsymbol{A}}$ with entries indexed in $\hat{S} \times \hat{S}$.*

| $(\lambda,\tau)$ | $(0.036, 10^{-10})$ | $(0.054, 0.01)$ | $(0.036, 0.01)$ | $(0.024, 0.1)$ |
|---|---|---|---|---|
| $\hat{\mathsf{df}}/n$ | $0.31 \pm 0.012$ | $0.21 \pm 0.0095$ | $0.3 \pm 0.011$ | $0.37 \pm 0.0093$ |
| $\hat{p}/n$ | $0.31 \pm 0.012$ | $0.22 \pm 0.0098$ | $0.31 \pm 0.012$ | $0.47 \pm 0.014$ |
| $\hat{n}/n$ | $0.83 \pm 0.011$ | $0.76 \pm 0.014$ | $0.83 \pm 0.012$ | $0.84 \pm 0.012$ |
| $\operatorname{tr}[\boldsymbol{\Sigma}\boldsymbol{A}]$ | $0.58 \pm 0.039$ | $0.39 \pm 0.027$ | $0.58 \pm 0.038$ | $0.8 \pm 0.038$ |
| $\|\operatorname{tr}[\boldsymbol{\Sigma}\boldsymbol{A}] - \hat{\mathsf{df}}/\operatorname{tr}[\boldsymbol{V}]\|$ | $0.0019 \pm 0.0015$ | $0.0015 \pm 0.0012$ | $0.0021 \pm 0.0016$ | $0.0023 \pm 0.0017$ |
| $\|\boldsymbol{\Sigma}^{1/2}(\widehat{\boldsymbol{\beta}} - \boldsymbol{\beta}^*)\|^2$ | $1.3 \pm 0.18$ | $1.7 \pm 0.25$ | $1.3 \pm 0.19$ | $1.9 \pm 0.21$ |
| $\zeta_1$ | $0.056 \pm 1$ | $0.021 \pm 1$ | $0.0044 \pm 1$ | $0.042 \pm 0.97$ |

Table 1: Simulation for Huber Elastic-Net regression under different choices of $(\lambda,\tau)$. $(n, p) = (1001, 1000)$. For each choice of $(\lambda,\tau)$, 600 data points are simulated with anisotropic design matrix and i.i.d. $t$-distributed noises with 2 degrees of freedom. A detailed setup is provided in Section 6.

253 The identities (19) are proved in [3, §2.6]. Simulations in Figures 1 to 3 and Table 1 illustrate typical
254 values for $\hat{\mathsf{df}}, \operatorname{tr}[\boldsymbol{V}], \operatorname{tr}[\boldsymbol{\Sigma}\widehat{\boldsymbol{A}}]$, the out-of-sample error and the criterion (2), $\hat{n} = \sum_{i=1}^n \psi'(r_i)$ and
255 $\hat{p} = |\hat{S}|$ under anisotropic Gaussian design and heavy-tailed $\varepsilon_i$. The simulation setup is as follows.

256 **Data Generation Process.** Simulation data are generated from a linear model $\boldsymbol{y} = \boldsymbol{X}\boldsymbol{\beta}^* + \boldsymbol{\varepsilon}$ with
257 anisotropic Gaussian design $\boldsymbol{\Sigma}$ and heavy-tail noise vector $\boldsymbol{\varepsilon}$. The design matrix $\boldsymbol{X}$ has $n = 1001$
258 rows and $p = 1000$ columns. Each row of $\boldsymbol{X}$ is i.i.d. $N(\boldsymbol{0}, \boldsymbol{\Sigma})$, with the same $\boldsymbol{\Sigma}$ across all
259 repetitions, generated once by $\boldsymbol{\Sigma} = \boldsymbol{R}^\top \boldsymbol{R}/(2p)$ with $\boldsymbol{R} \in \mathbb{R}^{2p \times p}$ being a Rademacher matrix with
260 i.i.d. entries $\mathbb{P}(\boldsymbol{R}_{ij} = \pm 1) = \frac{1}{2}$. The true signal vector $\boldsymbol{\beta}^* \in \mathbb{R}^p$ has its first 100 coordinates set to
261 $p^{1/2}/100 = \sqrt{10}/10$ and the rest 900 coordinates set to 0. The noise vector $\boldsymbol{\varepsilon} \in \mathbb{R}^n$ has i.i.d. entries
262 from the t-distribution with 2 degrees of freedom (so that $\operatorname{Var}[\varepsilon_i] = \infty$, i.e., $\varepsilon_i$ is heavy-tailed).

263 **Estimation Process.** Each dataset $(\boldsymbol{y}, \boldsymbol{X})$ is fitted by a Huber Elastic-Net estimator with
264 loss-penalty pair in (18). We focus on 2d heatmaps with respect to the two penalty parame-
265 ters $(\lambda,\tau)$ of the penalty; to this end the Huber loss parameter $\Lambda$ is set to $\Lambda = 0.054n^{1/2}$
266 and a grid for $(\lambda,\tau)$ in then set so that $\hat{\mathsf{df}}/n$ varies on the grid from 0 to 1 (cf. the mid-
267 dle heatmap in Figure 3). The Elastic-Net penalty $g(\boldsymbol{b};\lambda,\tau) = \lambda\|\boldsymbol{b}\|_1 + (\tau/2)\|\boldsymbol{b}\|_2^2$ is used
268 with $(\lambda,\tau) \in \{(0.036, 10^{-10}), (0.054, 0.01), (0.036, 0.01), (0.024, 0.1)\}$ in Figure 2 and Table 1,
269 $(\lambda,\tau) \in [0.0032, 0.41] \times \{10^{-10}\}$ in Figure 3, and $(\lambda,\tau) \in [0.0032, 0.041] \times [10^{-10}, 0.1]$ in Figure 1.
270 More simulation results are provided in the supplementary materials.

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
