# OpenReview forum: "Derivatives and residual distribution of regularized M-estimators with application to adaptive tuning"
_NeurIPS.cc/2021/Conference — NeurIPS 2021 Submitted_

### Official Review · Reviewer_eMyT · 2021-07-14

**Rating:** 5
**Confidence:** 2

**Summary:**

This paper studies M-estimators in linear models, with convex and gradient-Lipschitz loss function and a convex penalty.

Main contributions:
1. The authors provide a criterion to select tuning parameters (the loss function and the penalty function) of M-estimators, such that minimizing the criterion provides a proxy for minimizing the out-of-sample error, since the criterion is an approximation of the out-of-sample error up to a constant shift. One advantage of this criterion is that it does not require the prior knowledge of the covariance matrix of the design and the noise distribution.
2. The authors provide simulation results that confirm theoretical findings.

**Main Review:**

The paper is well organized. Some comments:

1. The criterion in Eq (2) exists for general $(\rho, g)$, while in the experiments it is only used for tuning the parameter with Huber loss and with Elastic-Net penalty. Some related questions on this: (1) If there exists an optimal estimator $\hat{\beta}$ that minimize the out-of-sample error? Does it has the form as Eq (1) for some $(\rho_0, g_0)$? (2) If the answers to (1) are both 'yes', is it possible to show that $(\rho_0, g_0)$ minimizes the criterion (over some space of $(\rho, g)$)?
2. The simulation part demonstrates how the criterion is calculated and its approximation accuracy of the out-of-sample error. One important missing part is the comparison with other parameter tuning methods: could this method outperform other parameter tuning methods?
3. The simulations in the appendix suggests that the results generalize to non-Gaussian design. What is the main technical difficulty of extending the results to non-Gaussian design?



**Time Spent Reviewing:**

5

---

> ### Author Response · Authors · 2021-08-10
> **Response to comments and questions by the referee**
>
> **Question 1**. The question whether there exists an optimal M-estimator of the form (1) for some $(\rho_0,g_0)$ is interesting, we are only aware of answers to that question in the special case where $\rho$ is the square loss.
> For the square loss $\rho(u)=u^2/2$, the question of the optimal convex penalty $g$ has been studied recently in Celentano and Montanari ("Fundamental Barriers to High-Dimensional Regression with Convex Penalties") when coefficients of $\beta^*$ are assumed to be generated iid from a prior distribution. In this square loss setting, the aforementioned paper shows that the answer is subtle: for certain prior distributions on the coefficients of $\beta^*$ some convex penalty $g_0$ achieves the best possible out-of-sample error among all known algorithms, while for other prior distributions there is a strict gap between all convex M-estimators and other non-convex algorithms (called Bayes-AMP in that paper). Beyond the square loss, whether convex $(\rho_0,g_0)$ exist that minimize the out-of-sample error is a delicate one that warrants further exploration.
>
> To answer the last question in item 1., if $(\rho,g)$ satisfies Assumptions 1.1-1.2, the techniques in the proof of Theorem 4.1 in fact shows that if $\epsilon_1,,...\epsilon_n$ are iid with finite variance, $\mathbb E[\|\Sigma^{1/2}(\hat\beta-\beta^*)\|^2 + \|\epsilon\|^2/n] = \mathbb E[\|r+\text{trace}[\Sigma\hat A]\psi(r)\|^2/n] + O(n^{-1/2})$, i.e., the approximation holds in expectation.
> Thus if $(\rho_0,g_0)$ and $(\rho_1,g_1)$ are two loss-penalty pairs, the one with the smallest expected out-of-sample error will have the smallest expected criterion up to a small $O(n^{-1/2})$ error term. Theorem 4.1 goes further, by showing that the approximation does not only hold in expectation but also in probability, which allows to control the approximation error between the out-of-sample error (plus $\|\epsilon\|^2/n)$ and the criterion simultaneously for several loss-penalty pairs, cf. for instance Corollaries 4.2 and 4.3. We will add a remark.
>
> **Question 2**. There are indeed other contenders for parameter tuning, most notably cross-validation. 5-fold Cross-validation, by design, picks an estimator which has smaller out-of-sample error using a subsample of size $4n/5$.
> In the challenging regime where $p/n\to$constant of interest here, picking the Lasso example for illustration (Bayati and Montanari, 2011) the out-of-sample error as a function of the tuning parameter $\lambda$ of the Lasso is given by the quantity $\tau_*(\lambda)$ found by solving a system of two equations as in Proposition 3.1 of Miolane and Montanari (2018). There, $n/p\to \delta$, and changing in the system $\delta$ for $4\delta/5$ (which is the biased-sample size corresponding to 5-Fold cross-validation) would significantly alter the out-of-sample curve $\tau_*(\lambda)$. On the other hand, the proposed criterion does not suffer this biased-sample size issue. For comparison, we will add the heatmaps for 2, 3, 5 and 10 fold cross-validation in the setting of Figure 1 to provide a comparison and to show the effect of the biased sample size.
>
> **Question 3**. Extending the theoretical results to non-Gaussian design is a major challenge, usually referred to as *universality*. The proofs here rely heavily on Stein formula, which is specific to Gaussian design. The other powerful tools that provide deep understanding of M-estimators in this challenging $p/n\to$constant regime are Approximate Message Passing and the Convex Gordon Min-max Theorem (Thrampoulidis et al.), both of which require Gaussian design. Although the limitation of all these techniques to Gaussian design has been known for a while, proving rigorous results appears to be a major challenge. There are a few papers in that direction (Montanari and Nguyen "Universality of the Elastic Net Squared Error"), but more general universality results are needed to rigorously bridge the gap between results for Gaussian design and their non-Gaussian counterparts. Such universality results would indeed be of major interest, but will require more than one conference paper.

---

### Official Review · Reviewer_ikDg · 2021-07-15

**Rating:** 6
**Confidence:** 3

**Summary:**

Finding the right parameter when doing robust estimation is a recurring problem in the robust community. This article propose to choose the parameters in a robust regularized linear regression model by minimizing a completely data-driven criterion. This criterion is shown to be a good approximation to the out of sample error, this is also illustrated on practical experiments.

**Limitations And Societal Impact:**

The main use for M-estimator is in robust statistics, however there are some limitations for the use of these results in robust statistics.  The article suppose Gaussian design matrix which is rather restrictive in practice and which shows that this article is about robustness in the target space but not in the feature space. A very important extension of this work would be when the design is heavy-tailed. Similarly, even though the error $\varepsilon$ can be anything as long as its distribution is continuous, there cannot be outliers because the outliers would change the $\beta^*$ and the criterion would then be non robust (for instance . This is a limitation for use in robust machine learning), this article is only valid for Gaussian design and uncorrupted, heavy-tailed target variable $y$.  This can be seen by the fact that the criterion depends on $r$ and if $y$ is corrupted or $X$ is corrupted, there can be some very large values of $r$.



**Main Review:**

The article is clear and well written, the results are very important in the robust machine learning community as it is often a problem to tune the estimators to a specific dataset. Most tuning procedures require knowledge on the variance of the inliers or a large computational effort (for instance through Lepski's method). In this article, the tuning is computationally tractable and the theory prove that this tuning is efficient with respect to the out of sample error.

**Time Spent Reviewing:**

3

---

> ### Author Response · Authors · 2021-08-10
> **Response to comments by Reviewer ikDg**
>
> Indeed, the proposed theory is only valid for Gaussian $X$ and uncorrupted, heavy-tailed noise; for instance results in Sections 4 and 5 assume that $\epsilon_i$ has $1+q$ moments for an arbitrarily small constant $q>0$.
> We note, however, that requiring only $1+q$ moments is an improvement over other theories developed in the $p/n\to$constant regime (for instance, Huang (2020, "Asymptotic risk and phase transition of L_1-penalized robust estimator") requires second moment of the noise, and the existence of the second moment is a recurring assumption for results obtained using Approximate Message Passing).
>
> In the present work, $p/n\to$constant and the goal is to capture the out-of-sample error, which is typically of order of a constant, up to a small additive error term. In particular there is no rate of convergence towards 0 (since out-of-sample errors are typically non-vanishing), and in this context we wish to compare the out-of-sample errors exactly as opposed to capturing a rate of convergence up to multiplicative constants.
> Previous works that allow adversarial outliers in the data and use Lepski's method for parameter tuning, e.g, Loh (2018) "Scale calibration for high-dimensional robust regression", try to capture the optimal rate of convergence towards 0 up to multiplicative constants. Lepski's method, by design, attempts to capture the optimal rate by allowing larger multiplicative constants (see the definition of the "majorant" in Goldenshluger and Lepski, 2008). Such parameter tuning schemes based on Lepski's method would not provide meaningful results in the context of interest in the submission where $p/n\to$constant and the out-of-sample errors are non-vanishing.
>
> We however agree that designs with heavy tails and adversarial contamination, in the context of $p\asymp n$ and non-vanishing errors, are of major interest for future work.

---

### Official Review · Reviewer_kzqs · 2021-07-16

**Rating:** 7
**Confidence:** 4

**Summary:**

The paper is dedicated to the analysis of robust linear regression, where the proposed estimation is based on sparse M-estimation. The statistical criterion corresponds to a penalised M-estimation problem, where both the non-penalised loss function and penalty function are assumed convex. The contributions of the paper can be summarised as follows:
- the authors provide a criterion for the optimal selection of the non-penalised loss and penalty function so that some out-of-sample distance, say D, is minimal.
- the authors quantify the behaviour of the penalised estimator \hat\beta using quantities depending on the observations only (idest no unknown quantities such as true sparse support) in a regime where p/n has finite limit.
- some explicit derivative formulas of the penalised estimator are provided.
- the distribution of the estimated residuals is provided, which allows for a selection of the tuning parameter for regularization.
- the authors derive an approximation formula of the distance D.


**Limitations And Societal Impact:**

I think the authors proposed a interesting way to specify a criterion for loss/penalty functions and their method is supported by some theoretical results. Please see my comments above concerning the limitations.

**Main Review:**

As a general comment, the idea of proposing a criterion for an optimal choice of both penalty and loss functions is interesting. There was an effort to derive theoretical results. That being said, I would have several concerns. I hope my comments will be useful.
- On the clarity:
1) I feel there is a significant lack of details supporting the underlying motivation and intuition of the proposed methodology. What is clearly the purpose? Do the authors aim at selecting a penalty function so that variable selection can be satisfied in a high-dimensional regime? Or rather than interpreting the model, do the authors aim at gaining prediction accuracy?
2) On the related works: the authors mention in the abstract "adaptive criterion to selection tuning parameters of penalised M-estimators". In the rest of the paper, I would have expected to find some theoretical results giving some explicit rates of \lambda to ensure the consistency of the penalised parameter. With this in mind, Loh's work (2017, "Statistical consistency and asymptotic normality for high-dimensional robust M-estimators") treated the problem of robust estimation with sparsity assumption and proposed a penalised M-estimation setting with non-convex penalisation. It would be relevant to discuss this further.
3) The simulation experiment seems like an afterthought and lacks further explanations: what properties of the proposed methodology are illustrated by the simulations?
- On the model assumptions: the key assumption is the convexity of both the loss function and the penalty function. The problem of convex penalty functions is that they do not provide good performances in terms of bias and variable selection. There are two ways to solve this issue: considering a two step procedure for incorporating stochastic adaptive weights in the penalty and alter the rate of \lambda; considering non-convex penalty functions as in Loh (2017). Should a non-convex penalty be chosen, the theoretical analysis might probably be altered. Would it be possible to consider non-convex penalty functions? How would it change the theoretical framework?
I am also wondering if this would be possible to relax the convexity of the loss function (which is not guaranteed especially in a high-dimensional regime) and assume alternative regularity conditions (such as restricted eigenvalue conditions).
- On the theoretical results:
The authors stated in Section 1.2 that their work is within the regime "p/n ---> finite limit". In assumption 1, the authors assume p/n <= \gamma, for \gamma >0. In theorem 3.2., a large sample distribution is provided when n --> \infty. How is the dimension p treated? Does the theorem hold when n,p --> \infty with the scaling behaviour p/n --> \gamma? It would be relevant to discuss this issue further.
In Theorem 3.1, the matrix \hat{A} is given by Theorem 2.1: there, a specific assumption is stated on \|\Sigma^1/2 \hat{A} \Sigma^1/2\|_op: such quantity depends on the dimension p; how does the inequality \|\Sigma^1/2 \hat{A} \Sigma^1/2\|_op <= 1/(n\mu) behave with respect to p?



**Time Spent Reviewing:**

Approximately 6.5 hours

---

> ### Author Response · Authors · 2021-08-10
> **Response to comments by Reviewer kzqs**
>
> **1**. *(...) lack of details supporting the underlying motivation*.
> The goal of the proposed observable criterion is to provide the ability to pick, between two or more convex regularized M-estimators, the one with the smallest out-of-sample error. Variable selection is not the goal and sparsity is not required for the theory, although the proposed methodology can be used in contexts where $\beta^*$ is sparse.
>
> **2** *(...) theoretical results giving some explicit rates of $\lambda$ to ensure consistency*.
> This is an interesting direction to pursue. The approach proposed in the paper is on the other hand more general, because it allows to compare estimators corresponding to unrelated loss-penalty pairs. For instance the proposed approach allows to compare an M-estimator obtained with Elastic-Net penalty to an M-estimator obtained with a Nuclear-norm penalty (viewing vectors in $R^p$ as matrices). In this context, consistency of the tuning parameter does not make sense because the two penalty functions are not proportional to each other. On a related note, studying consistency of the tuning parameter (towards an optimal parameter $\lambda^*$ that has smallest out-of-sample error) would require  further assumptions such as unicity of the optimal $\lambda^*$.
>
> **2**. *(...) Loh's work (2017 (...))*
> Thanks for the reference, we will add pointers to works on non-convex penalty functions. We note that this work tackles on sparsity settings with non-convex regularizers to improve upon the $\ell_1$ penalty, which is a very focused problem compared to the general approach proposed here (e.g., we allow general penalty functions).
>
>
> **3**. *(...) what properties of the proposed methodology are illustrated by the simulations?*
> Figure 1 illustrates that the approximation in Eq (3) (i.e., the validity of the proposed criterion) is correct on a wide 2-dimensional grid of tuning parameters. Figure 2 confirms the asymptotic normality proved in Theorem 3.1. Figure 3 confirms the findings of Theorem 5.1.
>
>
> $\bullet$ *The problem of convex penalty functions is that they do not provide good performances in terms of bias and variable selection*.
> As mentioned above, the submission is not focused on variable selection. We agree that there are multiple ways to fix the bias of certain convex methods; additionally to those mentioned in the referees report, de-biaisng (Zhang and Zhang, 2014), iterative hard thresholding, bootstrap.
> *Would it be possible to consider non-convex penalty functions? How would it change the theoretical framework?*
> This would be a very interesting avenue for future research. In the regime $p/n\to$constant of interest here, though, we are not aware of results for non-convex penalties. Since little is known about non-convex penalty in this regime, before getting to adaptive criterion of the out-of-sample error, a first step may be to generalize the system of equations of Bayati and Montanari (2011) that characterize the out-of-sample error of the Lasso, to non-convex penalty.
>
> $\bullet$ *I am also wondering if this would be possible to relax the convexity of the loss function (which is not guaranteed especially in a high-dimensional regime) and assume alternative regularity conditions (such as restricted eigenvalue conditions).*
> There is a misunderstanding here. When the submission mentions the convexity of the loss function, it refers to the convexity of the univariate function $\rho:R\to R$. For instance, the square loss $\rho(u)=u^2/2$ is always strongly convex, even though the mapping from $R^p\to R$ appearing in the objective function, $\beta\mapsto \sum_{i=1}^n \rho(y_i - x_i^T\beta)$, is not strongly convex when $p>n$.
>
> $\bullet$ *(...) How is the dimension p treated? Does the theorem hold when n,p --> \infty with the scaling behaviour p/n --> \gamma*
> All results hold as both $n,p\to+\infty$, while maintaining $p/n\le \gamma$. As explained between Assumption 1.1 and Assumption 1.2, by extracting a subsequence if necessary, there exists a constant $\gamma'\le\gamma$ such that $p/n\to\gamma'$ along this subsequence. The bound on $\|\Sigma^{1/2}\hat A\Sigma^{1/2}\|_{op}$ stated in Theorem 2.1 is not an assumption, but a property of the matrix $\hat A$ that is proved in Theorem 2.1. The matrix $\hat A$ is guaranteed to satisfy this bound under Assumption 1.1. Hence there is no hidden extra assumption on the growth of $p$.

---

### Official Review · Reviewer_A6hk · 2021-07-16

**Rating:** 6
**Confidence:** 2

**Summary:**

This paper studies the derivatives of regularized M-estimators and distribution of the residual.
These results give rises to a new novel adaptive criterion for tuning loss and penalty function.
Numerical experiments are conducted on various models to validate the theoretical result.

**Limitations And Societal Impact:**

This article clearly describes the assumptions needed for each theoretical result.

**Main Review:**

This paper is well-written and mostly clear.
The theory and method in this paper has many benefits, including general loss and penalty, $\Sigma$ and $\varepsilon_i$ agnostic criterion.
I am not an expert in this area and cannot judge the novelty and significance.

Question:
1. In the Theorem 3.1, it says $Z_i\sim N(0,1)$, but the dependency between $Z_i$ and $X$ is not clarified. Is it better to say  $(X_1,...,X_n,Z_i)\sim \otimes_{i=1}^n N(0,\Sigma)\otimes N(0,1)$ to explicitly show the dependency?
2. In line 417, how Proposition 7.1 is applied in the proof?

UPDATE: I thank the authors for addressing my questions and I decided to keep my score.

**Time Spent Reviewing:**

1 day

---

> ### Author Response · Authors · 2021-08-10
> **Point-by-point response to referees**
>
> 1. Thanks. We will clarify that $Z_i$ and the design matrix are **not** independent, but that $Z_i$ is independent of $\epsilon$ and of the $(n-1)$ rows $\\{\mathbf{x}_{l}, {l=1,...,n: l\ne i}\\}$ of the design matrix.
>
> 1. Proposition 7.1 is applied conditionally on the noise vector $\epsilon$ and on all the $(n-1)$ rows $\\{\mathbf{x}_{l}, {l=1,...,n: l\ne i}\\}$ of the design matrix.

---

### Official Review · Reviewer_nTec · 2021-07-17

**Rating:** 8
**Confidence:** 3

**Summary:**

The authors study M-estimators with convex penalties. They give algebraic forms for derivatives of the estimator with respect to response $y$ and the design matrix $X$. They describe the distribution of residuals or certain functions of residuals. Given a loss function and penalty (with tuning parameters) they propose a criterion that can effectively approximate the out-of-sample error. This criterion can be computed without the knowledge of the covariance matrix $\Sigma$ (rows of $X \sim N(0,\Sigma)$). They give finite sample bounds that show how fast the criterion converges to the out-of-sample error.



**Limitations And Societal Impact:**

I cannot see any negative societal impact due to this work.

**Main Review:**

The main results are all interesting. The manuscript sets up and explains the results in good detail.

Following are some limitations of the work. The design matrix $X$ is assumed to have Gaussian rows. This may not be true in practice, and it is not clear how good the criterion is without this assumption. The penalty $-\beta^T \Sigma \beta$ is assumed to be strongly convex. Therefore the results are not applicable when the estimator uses $\ell_1$ type of penalties.

Even with these limitations, the contributions are significant.


**Time Spent Reviewing:**

3

---

> ### Author Response · Authors · 2021-08-16
> **Answer to comments by Reviewer nTec**
>
> Thanks for your comments. We apologize for the delay in responding, and hope that the following will be informative/useful.
>
> **On the Gaussian assumption.** We agree that the Gaussian design assumption is a current limitation of
> the theory. Since the Gaussian assumption is central to our arguments, this is
> not a limitation that we can fix for the camera-ready version of the paper
> (however, note that simulations in the supplement for Rademacher design suggest
> that the theory applies more broadly; see also our response to Reviewer eMyT for
> a remark about universality and the related challenges).
>
> **On the strong convexity assumption.** On the other hand, the strong convexity assumption on the penalty is not central to the argument. We can for instance prove that similar theorems
> hold for the Huber loss with $\ell_1$-penalty, when the sparsity of $\beta^*$
> is not too large (e.g., coefficients of $\beta^*$ are iid from a distribution
> with a small enough point-mass at 0).
>
> A short version of the story is this. The main role of the strong convexity
> assumption is to provide the bound on the operator norm of $\hat A$ in Theorem
> 2.1. Without strong convexity, for the Huber loss with $\ell_1$-penalty and
> under sparsity assumption, the matrix $\hat A$ can also be bounded from above
> in operator norm, though with high-probability and not almost surely. This adds
> technicalities that need to be carefully taken care of with slightly different proofs, but it means that it is possible to extend our theorems to the
> $\ell_1$ penalty.
>
> The advantage of the strong convexity assumption of the
> initial submission is that it allows us to bypass these technicalities while
> obtaining general theorems (general in the sense that Assumption 1.1 covers
> *all* strongly convex penalties, it does not require separability, symmetry or
> any smoothness). The case of $\ell_1$ penalty was not included due to page
> constraints, and the rationale for excluding such results initially was that the
> challenges of the $\ell_1$ penalty had been solved before in related contexts
> (e.g., [2, 3, 7, 12, 15] in the present $p/n\to$constant regime, or in the $p>>>n$
> regime in works on the RE or RSC conditions by Bickel, Ritov and Tsybakov
> (2009); Negahban, Ravikumar, Wainwright, Yu (2010)). Because of these
> works, success at solving those challenges here for the $\ell_1$ penalty would not be *that* surprising
> (at least to us) compared to the theorems included in the submission which presents completely novel objects such as the new criterion in equation (2).
>
> **As accepted papers are allowed 1 extra page of content,** we will use this extra
> page to lay out a technical argument suitable for the $\ell_1$ penalty, and
> state the corresponding theorems (with rigorous proof in the supplement), for
> the Huber loss with $\ell_1$-penalty. This will illustrate that strong
> convexity is a convenient assumption that lets us bypass technicalities to
> present the general theory, but that the theory holds more broadly since those technicalities can be handled on a
> case-by-case basis for instance for the $\ell_1$ penalty.

---

### Comment · Area_Chair_SSGv · 2021-09-07
**Comparison with related work**

I believe the work of the authors has strong similarities with some recent works on the problem of risk estimation. In particular,

1. Dobriban, Wager, "High-dimensional asymptotics of prediction: Ridge regression and classification," Annals of Stat, 2018.

2. Rad, Maleki, "A scalable estimate of the extra-sample prediction error via approximate leave-one-out," JRSS-B,2020.

3. Rad, Zhou, Maleki, "Error bounds in estimating the out-of-sample prediction error using leave-one-out cross-validation in high-dimensions," AISTAT, 2020.

It seems that the discussions of these papers and a few more that study the risk estimation in this setting are completely missing in the paper. Any comment from the authors would help us in finalizing our decision.

---

> ### Author Response · Authors · 2021-09-08
> **Answer to AC comment**
>
> Dear AC and reviewers,
> Thanks for bringing these works to our attention and giving us an opportunity
> to compare these results with the submission.
>
>
>
> ### [1] Dobriban, Wager, "High-dimensional asymptotics of prediction: Ridge regression and classification," Annals of Stat, 2018.
>
> The intersection of [1] with our setting is for the square loss $\rho(u)=u^2/2$
> and Ridge penalty function $g(b)=\alpha\|b\|^2/2$ with a general, anisotropic
> covariance $\Sigma$ for the rows of the design.
> The covariance $\Sigma$ is assumed to have a limiting spectral distribution,
> and Theorem 2.1 in [1] characterizes the predictive risk
> $\|\Sigma^{1/2}(\hat\beta-\beta^*)\|^2$ of this Ridge regression problem with
> the companion Stieltjes transform $v(z)$ of the limiting spectral distribution
> of $\Sigma$: the limit of the predictive risk is given by
> $$\frac{1}{\lambda
>     v(-\lambda)}
>     \Bigl[1+\Bigl(\frac{\lambda\alpha^2}{\gamma}-1\Bigr)\Bigl(1-\frac{\lambda
>     v'(-\lambda)}{v(-\lambda)}\Bigr)
>     \Bigr]$$
> where $\alpha^2$ is the signal strength.
> The major difference with our work is that Theorem 2.1 from [1] provides
> characterization of the risk as a deterministic number that is typically
> unknown: it depends on several unknown quantity including the signal strength
> $\alpha^2$ and the limiting spectral distribution of $\Sigma$ through
> $v(-\lambda),v'(-\lambda)$.
> The main results of present submission are of a very different nature: it aims
> to provide a fully data-driven criterion, that does not depend on any of those
> typically unknown quantities, to estimate
> $\|\Sigma^{1/2}(\hat\beta-\beta^*)\|^2$.
> Our result is thus more applicable (the risk can be estimated directly from the
> data, no need to know the companion Stieltjes transform)
> and has broader scope as any strongly-convex penalty $g$ is allowed,
> including non-smooth ones.
>
> The different nature can also be seen in the proofs: we do not rely at all on
> random matrix theory results (Marcenko-Pastur; Bai and Silverstein, 2010), for
> instance this lets us handle random matrices with a random, dependent dimension
> (e.g., matrix $\hat A$ in the Elastic-Net case has rank $|\hat S|$ and this
> rank itself depends on $X$) for which no known random matrix theory result
> applies.
>
> As the AC comment mentions that we may have missed other related results, we
> note in passing that results arising from Approximate Message Passing, or the
> Convex Gordon Min-Max theorem, also characterize the predictive risk as a
> deterministic limit that depends on unknown quantities (e.g., solving the
> system of 2 equations in Bayati and Montanari (2011), Miolane and Montanari
> (2018) requires the knowledge of a prior on $\beta^*$ which is typically
> unknown).
> The criterion we propose in equation (2), on the other hand, can be fully computed from the data.
>
>
> ### [2] Rad, Maleki, "A scalable estimate of the extra-sample prediction error via approximate leave-one-out," JRSS-B,2020.
>
> [2] develops a so-called Approximate Leave-one Out (ALO) criterion to estimate the out-of-sample error
> for different metrics $\phi$.
> The goal of the ALO is similar to the goal of our criterion (2), that is,
> to estimate the out-of-sample error using observable quantities.
> For comparison, take $\phi(u,v)=(u-v)^2/2$ in [2] so that the target of
> estimation (out-of-sample error) is the same target as the present submission.
>
> A major improvement of the theory of the submission compared to [2]
> is that we provide an end-to-end analysis of non-smooth penalty functions
> (as well as losses $\rho$ with $\rho''$ such as the Huber loss).
> Our statistical guarantees in Sections 3, 4 and 5 apply to any strongly convex
> penalty with no restriction on its smoothness, e.g.,
> $g(b)=\tau\|b\|_2^2/2 + \lambda \|b\|_1$ (Elastic-Net),or $g(b)=\tau\|b\|_2^2/2 + \lambda \|b\|_\{nuclear\}$
> (Ridge plus Nuclear norm, when viewing $\mathbb R^p$ as a space of matrices $\mathbb R^{d_1\times d_2}$
> with $p=d_1d_2$).
> This is because the theoretical analysis is carried out directly for the M-estimator
> of interest, without resorting to an approximation argument.
>
> On the other hand, attempting to find comparable theoretical results in [2], we find that [2] proves the following results:
>
> - [2, Theorem 1] which shows that if a non-smooth penalty $r(b)$
> is replaced with a smoothed version $r^\alpha(b)$,
> then as the smoothing parameter $\alpha$ tends to infinity
> $\lim_{\alpha\to+\infty} ALO^\alpha = ALO$, i.e., the ALO criterion
> of the M-estimator with smoothed penalty converges to the ALO criterion
> of the original M-estimator. This approximation argument is valid for Lasso,
> Elastic-Net and Bridge penalty functions and is completely deterministic. (It is unclear, though, how to extend this to other non-smooth penalty functions for instance ones involving the nuclear norm).
>
> - Statistical guarantees for M-estimators with twice-differentiable
> penalty functions satisfying [2, Assumption 6] are given
> in [2, Theorem 2 and Corollary 1]. Assuming that the computationally
> costly
> Leave-One-Out correctly estimates the out-of-sample estimate ([3] below offers a justification for this),
> this is the closest result of [2] to our theoretical guarantees.
> [2, Theorem 2] requires in Assumption 6 an upper bound $c_2(n)$ on some form
> of Lipschitz condition for the second derivatives of the penalty
> (specifically, cf. Eq. (24) in [2]), and
> [2, Lemma 2] goes on to show that $c_2(n) \le 4\alpha^2$ for the $\alpha$-smoothed
> L1 penalty and $\alpha$-smoothed Elastic-Net penalty.
> The bound in [2, Theorem 3] is vacuous if $c_2(n)>>n^{1/4}$.
>
> However, in order to obtain statistical guarantees and
> a complete end-to-end analysis of an M-estimator
> with non-smooth penalty, for instance the Elastic-Net, one still needs
> to balance the two bounds. The second item above requires $4\alpha^2=c_2(n)=o(n^{1/4})$ while the first item above requires $\alpha\to+\infty$,
> and no non-asymptotic bounds are available
> for the approximation $ALO^\alpha \approx ALO$. This prevents from
> balancing the two phenomena and obtaining, for non-smooth penalty functions, statistical guarantees for ALO
> similar to [2, Theorem 3] or those derived in the submission in Sections 3, 4, 5.
> Looking at more recent works that cite [2], it appears that this issue has not
> yet been fixed and it is unknown if [2, Theorem 3] is mathematically valid for non-smooth penalty functions.
>
> For the loss function, a similar improvement is provided by the submission as
> it can handle the Huber loss (whose second derivative is not continuous), while
> Assumption 6 (through Eq. (23)) in [2] requires some form of Holder smoothness
> of the second derivative.
>
> We are very grateful for bringing this line of work to our attention as it is clearly
> related, and the ALO criterion can be compared in simulations to our criterion in equation (2);
> we will discuss these works in the camera-ready version if the submission is accepted.
> However, as explained above, the statistical guarantees offered in Sections 3, 4 and 5
> of the submission for *non-smooth*
> penalty functions have no comparable results in [2] unless [2, Theorem 3 and Corollary 1] are
> stretched outside of their formal setting and assumptions. We haven't been able to locate
> publicly available results similar to [2, Theorem 2 or Corollary 1] applicable
> to the Elastic-Net penalty for instance.
>
> ### [3] Rad, Zhou, Maleki, "Error bounds in estimating the out-of-sample prediction error using leave-one-out cross-validation in high-dimensions," AISTAT, 2020.
>
> The paper [3] provides the rational for using the leave-one-out estimate as a proxy
> for the out-of-sample error. This justifies the approach in [2].
> The submission on the other hand does not rely on the closeness of the leave-one-out estimate
> to the out-of-sample error. This may explain that our "direct" approach is
> what allows to handle non-smooth penalty and go beyond Assumption 6 in [2].

---

### Decision · Program_Chairs · 2021-09-27

**Decision:**

Reject

**Comment:**

All the reviewers agree that the paper makes interesting contributions to the field of high-dimensional statitics. The authors provided informative answers to the questions raised by the area chair and the reviewers as well. The area chair has a positive view of the paper as well and believes that the paper should be eventually published at Neurips or similar venues. However, the paper suffers from a weakness; It missed some of the related work that seemed quite related to the contributions of the paper. The authors provided interesting conceptual comparisons in their response to the reviewers' comments. However, for the following two reasons the paper is rated below the other papers presented at Neurips:

1. A proper and more detailed comparison with related work should be part of the original paper so that all the reviewers can judge the paper better.

2. A more quantitative and numeric comparison is required to help reviewers see how the paper's contributions are compared with the existing methods.

Based on these issues, the reviewers and the area chair have rated this paper slightly below the other papers that have been accepted at Neurips. However, we would like to encourage the authors to submit this paper to future venues. It seems that the paper can become a major contribution to the field of high-dimensional statistics.